# BEEF: Bi-Compatible Class-Incremental Learning via Energy-Based Expansion and Fusion

**Fu-Yun Wang**[1,2]*  **Da-Wei Zhou**[1]  **Liu Liu**[2]†  **Han-Jia Ye**[1]†  **Yatao Bian**[2]†
**De-Chuan Zhan**[1]  **Peilin Zhao**[2]
[1]State Key Laboratory for Novel Software Technology, Nanjing University  [2]Tencent AI Lab
wangfuyun@smail.nju.edu.cn  {zhoudw,yehj,zhandc}@lamda.nju.edu.cn
{leonliuliu, yataobian}@tencent.com  peilinzhao@hotmail.com

## Abstract

Neural networks suffer from catastrophic forgetting when sequentially learning tasks phase-by-phase, making them inapplicable in dynamically updated systems. Class-incremental learning (CIL) aims to enable neural networks to learn different categories at multi-stages. Recently, dynamic-structure-based CIL methods achieve remarkable performance. However, these methods train all modules in a coupled manner and do not consider possible conflicts among modules, resulting in spoilage of eventual predictions. In this work, we propose a unifying energy-based theory and framework called **B**i-Compatible **E**nergy-Based **E**xpansion and **F**usion (BEEF) to analyze and achieve the goal of CIL. We demonstrate the possibility of training independent modules in a decoupled manner while achieving bi-directional compatibility among modules through two additionally allocated prototypes, and then integrating them into a unifying classifier with minimal cost. Furthermore, BEEF extends the exemplar-set to a more challenging setting, where exemplars are randomly selected and imbalanced, and maintains its performance when prior methods fail dramatically. Extensive experiments on three widely used benchmarks: CIFAR-100, ImageNet-100, and ImageNet-1000 demonstrate that BEEF achieves state-of-the-art performance in both the ordinary and challenging CIL settings. The Code is available at https://github.com/G-U-N/ICLR23-BEEF.

## 1 Introduction

The ability to continuously acquire new knowledge is necessary in our ever-changing world and is considered a crucial aspect of human intelligence. In the realm of applicable AI systems, it is expected that these systems can learn new concepts in a stream while retaining knowledge of previously learned concepts. However, deep neural network-based systems, which have achieved great success, face a well-known issue known as catastrophic forgetting (French, 1999; Golab & Özsu, 2003; Zhou et al., 2023b), whereby they abruptly forget prior knowledge when directly fine-tuned on new tasks. To address this challenge, the class-incremental learning (CIL) field aims to design learning paradigms that enable deep neural networks to learn novel categories in multi-stages while maintaining discrimination abilities for previous ones (Rebuffi et al., 2017; Zhou et al., 2023a).

Numerous approaches have been proposed to achieve the goal of CIL, with typical methods falling into two groups: regularization-based methods and dynamic-structure-based methods. Regularization-based methods (Kirkpatrick et al., 2017; Aljundi et al., 2018; Li & Hoiem, 2017; Rebuffi et al., 2017) add constraints (*e.g.*, parameter drift penalty) when updating, thus forcing the model to maintain crucial information for old categories. However, these methods often suffer from the stability-plasticity dilemma, lacking the capacity to handle all categories simultaneously. Dynamic-structure-based methods (Yan et al., 2021; Li et al., 2021) expand new modules at each learning stage to enhance the model's capacity and learn the task-specific knowledge through the new module, achieving remarkable performance. Whereas, these methods have an intrinsic drawback. They

---

*This work is done when Fu-Yun Wang works as an intern in Tencent AI Lab.
†Correspondence to: Han-Jia Ye (yehj@lamda.nju.edu.cn), Liu Liu (leonliuliu@tencent.com), Yatao Bian (yataobian@tencent.com)

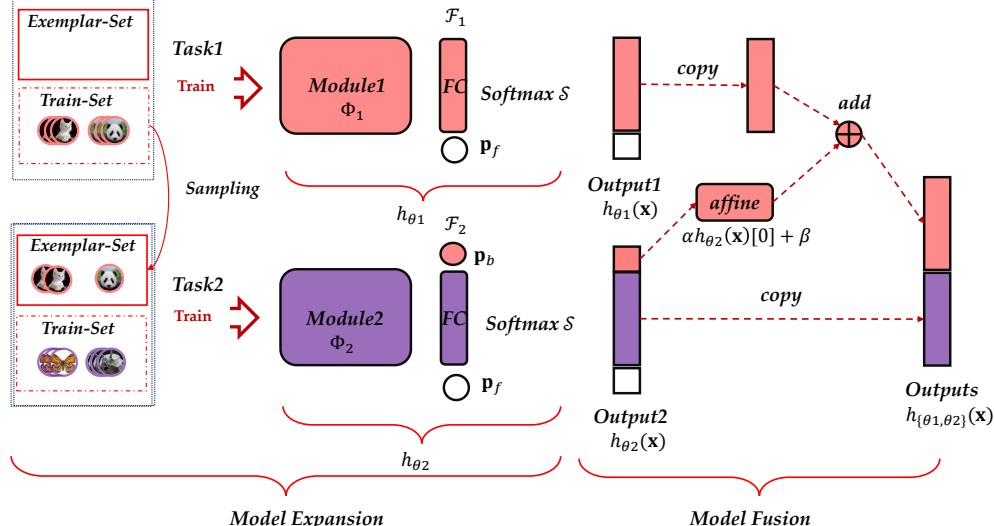

Figure 1: **The conceptual illustration of BEEF.** The training consists of two phases: expansion and fusion. At the expansion phase, we *independently* train the new module for the current task, while classifying all the samples from prior tasks into $\mathbf{p}_b$ and classifying all samples generated from the built-in energy-based model into $\mathbf{p}_f$. At the fusion phase, the output of $\mathbf{p}_b$ is equally added to the output of prior modules to mitigate the task-bias and form a unifying classifier.

directly retain all learned modules without considering conflicts among modules, thus corrupting the joint feature representation and misleading the ultimate predictions. For example, old modules not considering the future possible updates may mislead the final prediction for new classes. This defect limits their performance on long-term incremental tasks.

In this paper, we aim to reduce the possible conflicts in dynamic-structure-based methods and innovatively propose to achieve *bi-directional compatibility*, which consists of backward compatibility and forward compatibility. To be specific, *backward compatibility* is committed to making the discrimination ability of old modules unaffected by new ones. On the contrary, *forward compatibility* aims to reduce impact of old modules on the ultimate predictions when new categories emerge. By achieving bi-directional compatibility, given a sample from a specific task, the module responsible for the corresponding task will dominate the ultimate predictions in the ideal case.

Fig. 1 displays the BEEF training framework, which is made of two phases (Model Expansion and Model Fusion). At the expansion phase, we assume that different modules are independent and train them isolatedly. Then, at the fusion phase, all trained modules are combined to form a unifying classifier. To achieve bi-directional compatibility, we introduce two additional prototypes: forward prototype $\mathbf{p}_f$ and backward prototype $\mathbf{p}_b$. The backward prototype $\mathbf{p}_b$ is set to measure the confidence of old classes, while the forward prototype $\mathbf{p}_f$ aims to measure uncertainty in the open world. To be specific, when training a new module, we set $\mathbf{p}_b$ as the cluster prototype for all old classes and use it to learn a task boundary between the current task and prior ones. In the meanwhile, we set $\mathbf{p}_f$ as the cluster prototype for samples from the unseen distributions generated through energy-based sampling and thus use it to measure uncertainty and better capture the current distribution. We show that BEEF can be deduced from a unifying extendable energy-based theoretical framework, which allows us to transform the open-world problem into a normal classification problem and model the input distribution synchronously while learning the discrimination ability. Vast experiments on three widely-used benchmarks show that our method achieves state-of-the-art performance. Besides, prevalent CIL methods all require a well-selected balanced exemplar-set for rehearsal, which might be impractical due to data privacy issues (Delange et al., 2021; Ji et al., 2014) and computation cost when choosing exemplars. Our method pushes it to a harder setting. With only randomly sampled data from prior tasks, BEEF maintains its effectiveness while the performance of other methods declines drastically.

## 2 RELATED WORK

**Incremental learning.** Most recent studies on incremental learning are either task-based or class-based. The crucial difference between them is whether task-id is known at the evaluation phase (Van de

Ven & Tolias, 2019). Our work is a class-based method, which is typically called class-incremental learning (CIL). Prevalent CIL methods can be categorized into two classes: regularization-based, and dynamic-structure-based. *Regularization-based* methods impose constraints when learning tasks. Kirkpatrick et al. (2017); Aljundi et al. (2018) penalize the parameter drift. Li & Hoiem (2017); Rebuffi et al. (2017); Wu et al. (2019); Zhao et al. (2020) utilize knowledge distillation (Hinton et al., 2015) to constrain the model's output. Douillard et al. (2020) propose a novel spatial knowledge distillation. Zhou et al. (2022) propose the concept of *forward compatible* (Gheorghioiu et al., 2003; Shen et al., 2020) and squeeze the space for known categories, thereby reserving feature space for future categories. *Dynamic-structure-based* methods create new modules to enhance the capacity for learning new tasks. Yan et al. (2021); Li et al. (2021) combine all modules together to form a unifying classifier, but it leads to an increasing training cost. Douillard et al. (2021) applies transformer (Dosovitskiy et al., 2020; Touvron et al., 2021) to CIL and dynamically expands task tokens when learning new tasks. Wang et al. (2022a) proposes to dynamically expand and compress the model based on gradient boosting (Mason et al., 1999) to adaptively learn new tasks. Liu et al. (2021b; 2023) cleverly apply reinforcement learning in CIL to obtain univserally better memory management strategy or hyperparameters. However, prevalent CIL approaches usually require a well-selected class-balanced exemplar-set for rehearsal (Rebuffi et al., 2017), which has an evident impact on their performance (Masana et al., 2020) as we verify experimentally. BEEF not only achieves state-of-the-art performance but shows strong robustness to the choice of exemplar-set.

**Energy-based learning.** EBMs define probability distributions with density proportional to $\exp(-E)$, where $E$ is the energy function (LeCun et al., 2006). So far, the theory and implementation of EBMs have been well studied. Xie et al. (2016) show that the generative random field model can be derived from the discriminative ConvNet. Xie et al. (2018a; 2022) well study the cooperative training of two generative models for image modeling and synthesis. Additionally, Nijkamp et al. (2019) propose to treat the non-convergent short-run MCMC as a learned generator model or a flow model and show that the model is capable of generating realistic samples. Xie et al. (2021c) propose to learn a VAE to initialize the finite-step MCMC for efficient amortized sampling of the EBM. Xiao et al. (2021) propose a symbiotic composition of a VAE and an EBM that can generate high-quality images while achieving fast traversal of the data manifold. Zhao et al. (2021) propose a multistage coarse-to-fine expanding and sampling strategy, which starts with learning a coarse-level EBM from images at low resolution and then gradually transits to learn a finer-level EBM from images at higher resolution. Besides, EBMs have been successfully applied in many fields such as data generation (Zhai et al., 2016; Zhao et al., 2016; Deng et al., 2020; Du & Mordatch, 2019) with various data formats including graph (Liu et al., 2021a), video (Xie et al., 2017; 2019), 3D volumetric shape (Xie et al., 2018b; 2020), 3D unordered point cloud (Xie et al., 2021a), image-to-image translation (Xie et al., 2021c;b), saliency map (Zhang et al., 2022), *etc.*, out-of-distribution detection (OOD) (Hendrycks & Gimpel, 2016; Bai et al., 2021; Liu et al., 2020; Lee et al., 2020; Lin et al., 2021), and density estimation (Silverman, 2018; Zhao et al., 2016), *etc*. Wang et al. (2021) model the open world uncertainty as an extra dimension in the classifier, achieving better calibration in OOD datasets. Grathwohl et al. (2019) propose to model the joint distribution $\mathbb{P}(\mathbf{x}, y)$ for the classification problem. Bian et al. (2022) apply energy-based learning for cooperative games and derive new player valuation methods. Zheng et al. (2021) propose to represent the statistical distribution within a single natural image through an EBM framework. Xu et al. (2022) apply EBM for inverse optimal control and autonomous driving. There have been several attempts to apply EBMs to incremental learning. Li et al. (2020) propose a novel energy-based classification loss and network structure for continual learning. Joseph et al. (2022) build an energy-based latent aligner that recovers the corrupted latent representation. Wang et al. (2022b) propose anchor-based energy self-normalization classifier in incremental learning.

## 3 METHOD

In this section, we give a description of BEEF and how we apply EBM to CIL to learn a unifying classifier while achieving bi-directional compatibility. In Sec. 3.1, we first introduce some basic knowledge of CIL. In Sec. 3.2, we present the definition of energy and then deduce optimization objective at the expansion phase. Then, to avoid the intractability of normalizing constant, we prove a gradient equivalent objective and explain why it helps to achieve the bi-directional ability. After that, we propose an efficient yet effective fusion strategy in Sec. 3.3. Additionally, we extend BEEF to Stable-BEEF by allocating multiple forward prototypes and backward prototypes. The detailed illustration and proof are provided in Appendix B.

## 3.1 PRELIMINARIES

CIL aims to learn a unifying classifier from a sequence of data divided into several incremental sessions with different class groups. At the $t^{th}$ incremental session, the model receives the training dataset $\mathcal{D}_t = \{(\mathbf{x}_i^t, y_i^t)\}_{t=1}^N$, where $\mathbf{x}_i^t \in \mathcal{X}_t$ is an input sample and $y_i^t \in \mathcal{Y}_t$ is the corresponding label, which is not accessible at latter sessions. Only a small amount of exemplars of previous categories are retained in a size-limited exemplar-set $\mathcal{V}_t \subseteq \cup_{i=1}^{t-1}\mathcal{D}_i$. The model is expected to train on $\mathcal{D}_t \cup \mathcal{V}_t$ and be evaluated on the test set of all known categories. At the following discussions, we focus on the details of the $t^{th}$ incremental session without loss of generality. Particularly, we denote the label spaces of all known classes and novel classes as $\mathcal{Y}_o = \cup_{i=1}^{t-1}\mathcal{Y}_i$ and $\mathcal{Y}_n = \mathcal{Y}_t$, respectively. $|\mathcal{Y}_n| = K$ and $|\mathcal{Y}_o| = M$, representing the number of new categories and that of old ones.

## 3.2 ENERGY-BASESD MODEL EXPANSION

Let $h_\theta : \mathcal{X} \longrightarrow \Delta^{K+1}$ be the newly created module (typically as a single-skeleton CNN), where $\mathcal{X} = \cup_{i=1}^t \mathcal{X}_i$ and $\Delta^{K+1}$ is a $K + 1$-standard simplex (*i.e.*, $K + 2$ dimensional vectors with non-negative elements that sum up to 1). Therefore, we can further decompose $h_\theta$ as $\mathcal{S} \circ \mathcal{F} \circ \Phi$, where $\Phi : \mathcal{X} \longrightarrow \mathbb{R}^d$ is the non-linear feature extractor, $\mathcal{F} : \mathbb{R}^d \longrightarrow \mathbb{R}^{K+2}$ is a linear classifier transforming the feature into the $K + 2$ - dimensional logits, and $\mathcal{S}$ denotes the non-linear activation function softmax which constrains the final output on the $K + 1$- standard simplex. $h_\theta(\mathbf{x})[k]$ represents the $k + 1^{th}$ element of the final output. Ignoring the bias, $\mathcal{F}$ can be denote as a $d \times (K + 2)$ matrix $\mathbf{F} = [\ \mathbf{p}_b \quad \mathbf{F}_{\text{base}} \quad \mathbf{p}_f \ ]$, where $\mathbf{F}_{\text{base}}$ with shape $d \times K$ is the base classifier for the current task, and $\mathbf{p}_b$ / $\mathbf{p}_f$ is an additional prototype that measures *past confidence* / *future uncertainty*, transforming the feature extracted from $\Phi$ into logits at index $0$ / $K + 1$.

First, given an input-label pair $(\mathbf{x}, y) \in \mathcal{X} \times (\mathcal{Y}_o \cup \mathcal{Y}_n)$, we define the energy $E_\theta(\mathbf{x}, y)$ as

$$E_\theta(\mathbf{x}, y) = \begin{cases} -\log h_\theta(\mathbf{x})[\sigma(y)], & y \in \mathcal{Y}_n \\ -\log (h_\theta(\mathbf{x})[0]/M), & y \in \mathcal{Y}_o \end{cases}, \tag{1}$$

where $\sigma : \mathcal{Y}_n \longrightarrow 1, 2, \ldots, K$ is a bijection function mapping a given label to its corresponding class index. The energy $E(\mathbf{x}, y)$ measures the uncertainty of predicting $\mathbf{x}$'s label as $y$. Hence, the definition of the energy is compatible with traditional classification definitions, since we typically use $h_\theta(\mathbf{x})[\sigma(y)]$, which is the negative exponent of the energy, to indicate the confidence of predicting $\mathbf{x}$'s label as $y$. Moreover, we use $h_\theta(\mathbf{x})[0]$ to represent the overall confidence of $\mathbf{x}$'s label belonging to $\mathcal{Y}_o$ and do not expect the new module to distinguish between the old categories. Hence, $E(\mathbf{x}, y)$ for any $y \in \mathcal{Y}_o$ is represented as $-\log (h_\theta(\mathbf{x})[0]/M)$. The denominator $M$ makes the energy larger and indicates that the current module has a larger uncertainty about old categories due to the limited supervision for old categories from $\mathcal{V}_t$.

Since $\mathbb{P}_\theta(y|\mathbf{x}) = \frac{\exp(-E_\theta(\mathbf{x},y))}{\sum_{y'} \exp(-E_\theta(\mathbf{x},y'))}$ and $\mathbb{P}_\theta(\mathbf{x}) = \frac{\sum_{y'} \exp(-E_\theta(\mathbf{x},y'))}{\sum_{\mathbf{x}'} \sum_{y'} \exp(-E_\theta(\mathbf{x}',y'))}$, the conditional probability density and marginal probability density can be formulated as

$$\mathbb{P}_\theta(y|\mathbf{x}) = \begin{cases} \frac{h_\theta(\mathbf{x})[\sigma(y)]}{\sum_{k=0}^K h_\theta(\mathbf{x})[k]}, & y \in \mathcal{Y}_n \\ \frac{h_\theta(\mathbf{x})[0]/M}{\sum_{k=0}^K h_\theta(\mathbf{x})[k]}, & y \in \mathcal{Y}_o \end{cases}, \qquad \mathbb{P}_\theta(\mathbf{x}) = \frac{\sum_{k=0}^K h_\theta(\mathbf{x})[k]}{\sum_{\mathbf{x}'} \sum_{k=0}^K h_\theta(\mathbf{x}')[k]}. \tag{2}$$

We define the energy function $E_\theta(\mathbf{x})$ via $\mathbb{P}_\theta(\mathbf{x}) = \frac{\exp(-E_\theta(\mathbf{x}))}{\sum_{\mathbf{x}'} \exp(-E_\theta(\mathbf{x}'))}$ , then $E_\theta(\mathbf{x})$ is formulated as

$$E_\theta(\mathbf{x}) = -\log \sum_{k=0}^K h_\theta(\mathbf{x})[k]. \tag{3}$$

With energy functions defined above, we give proof of the derivation of our optimization objective when training a new module and demonstrate how it works to achieve bi-directional compatibility. Instead of simply learning a discriminator $\mathbb{P}_\theta(y|\mathbf{x})$, which usually causes overconfident predicts even when receiving samples from unseen distributions, we estimate the joint distribution $\mathbb{P}_\theta(\mathbf{x}, y)$

$$\underset{\theta}{\arg\min} \quad \mathbb{E}_{\mathbb{P}_{real}(\mathbf{x},y)} \left[ -\log \mathbb{P}_\theta(\mathbf{x}, y) \right] \tag{4}$$

$$= \underset{\theta}{\arg\min} \quad \mathbb{E}_{\mathbb{P}_{real}(\mathbf{x})} \left[ -\log \mathbb{P}_\theta(\mathbf{x}) \right] + \mathbb{E}_{\mathbb{P}_{real}(\mathbf{x},y)} \left[ -\log \mathbb{P}_\theta(y|\mathbf{x}) \right]. \tag{5}$$

The estimation of the joint distribution not only encourages the model to learn to distinguish all known categories but also helps model the input distribution, thus making the model sensitive to the

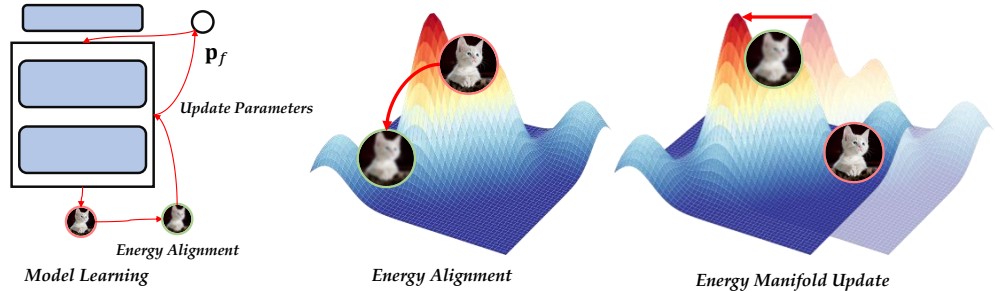

Figure 2: **Learning energy manifold.** The learning process of the energy manifold. The left part illustrates the concrete workflow of model training. The middle and right part illustrate the corresponding high-level abstract energy manifold learning process.

input distribution drift. Therefore, when unseen categories emerge, it helps those modules alleviate overconfident predictions and reduce their impact on the ultimate predictions. However, due to the intractability of the normalizing constant $\sum_{\mathbf{x}'} \sum_{y'} \exp(-E_\theta(\mathbf{x}', y'))$, we optimize the gradient equivalent objective of Eq. 4.

**Theorem 3.1** (Marginal Distribution Maximum Likelihood Estimation). *Defining $E'_\theta(\mathbf{x}) = -\log h_\theta(\mathbf{x})[K+1]$ and its corresponding marginal distribution as $\mathbb{P}'_\theta(\mathbf{x})$, the optimization of $\mathbb{E}_{\mathbb{P}_{real}(\mathbf{x})}[-\log \mathbb{P}_\theta(\mathbf{x})]$ is equivalent to that of $\mathbb{E}_{\mathbb{P}_{real}(\mathbf{x})}\left[-\log \sum_{k=0}^{K} h_\theta(\mathbf{x})[k]\right] + \lambda_{\bar{\theta}}\mathbb{E}_{\mathbb{P}'_{\bar{\theta}}(\mathbf{x})}[-\log h_\theta(\mathbf{x})[K+1]]$ when gradient descend is applied, where $\lambda_{\bar{\theta}}$ is the ratio of the normalizing constants determined by $E'_\theta(\mathbf{x})$ and $E_\theta(\mathbf{x})$, and $\bar{\theta}$ means that parameters of $\theta$ is frozen (i.e., instances sampled from $\mathbb{P}'_{\bar{\theta}}(\mathbf{x})$ are detached).*

**Theorem 3.2** (Conditional Distribution Maximum Likelihood Estimation). *With preliminaries from Thm. 3.1, the optimization of $\mathbb{E}_{\mathbb{P}_{real}(\mathbf{x},y)}[-\log \mathbb{P}_\theta(y|\mathbf{x})]$ is equivalent to that of $\mathbb{E}_{\mathbb{P}_{real}(\mathbf{x},y)}[-\log h_\theta(\mathbf{x})[\sigma'(y)]] + \mu_{\bar{\theta}}\mathbb{E}_{\mathbb{P}_{real}(\mathbf{x})}[-\log h_\theta(\mathbf{x})[K+1]]$ when gradient descend is applied, where $\mu_{\bar{\theta}} = \frac{h_{\bar{\theta}}(\mathbf{x})[K+1]}{\sum_{k=0}^{K} h_{\bar{\theta}}(\mathbf{x})[k]}$, $\sigma'(y) = \begin{cases} \sigma(y), & y \in \mathcal{Y}_n \\ 0, & y \in \mathcal{Y}_o \end{cases}$.*

Due to the space limit, detailed proofs for Thm. 3.1 and Thm. 3.2 are deferred to Appendix A. Combining Thm. 3.1 and Thm. 3.2, the ultimate optimization objective can be formulated as

$$\mathbb{E}_{\mathbb{P}_{real}(\mathbf{x})}\left[-\log \sum_{k=0}^{K} h_\theta(\mathbf{x})[k]\right] + \lambda_{\bar{\theta}}\mathbb{E}_{\mathbb{P}'_{\bar{\theta}}(\mathbf{x})}[-\log h_\theta(\mathbf{x})[K+1]] + \\ \mathbb{E}_{\mathbb{P}_{real}(\mathbf{x},y)}[-\log h_\theta(\mathbf{x})[\sigma'(y)]] + \mu_{\bar{\theta}}\mathbb{E}_{\mathbb{P}_{real}(\mathbf{x})}[-\log h_\theta(\mathbf{x})[K+1]] ,$$

(6)

which is upper bounded by

$$\text{(Objective)} \quad 2\mathbb{E}_{\mathbb{P}_{real}(\mathbf{x},y)}[-\log h_\theta(\mathbf{x})[\sigma'(y)]] + \mu_{\bar{\theta}}\mathbb{E}_{\mathbb{P}_{real}(\mathbf{x})}[-\log h_\theta(\mathbf{x})[K+1]] + \\ \lambda_{\bar{\theta}}\mathbb{E}_{\mathbb{P}'_{\bar{\theta}}(\mathbf{x})}[-\log h_\theta(\mathbf{x})[K+1]] .$$

(7)

We take Eq. 7 as the eventual training objective. Here, we explain the *roles of different components* and *the reason why better bi-directional compatibility is achieved* through optimizing this objective. $\mathbb{E}_{\mathbb{P}_{real}(\mathbf{x},y)}[-\log h_\theta(\mathbf{x})[\sigma'(y)]]$ prompts the new module to accurately discriminate all categories from current task as well as build explicit decision boundaries between current task and prior ones. By setting $\mathbf{p}_b$ as the special prototype for all old categories, we can better exploit the shared structure of all old categories and reduce the risk of over-fitting, which typically results from inadequate training samples on old categories. Given a sample from prior tasks, the new module perceives this task boundary and reduces the confidence of its own task, so that the old module dominates the ultimate prediction. Hence, we achieve better *backward compatibility* for old categories than naively tuning the new module on all categories. $\mathbb{E}_{\mathbb{P}_{real}(\mathbf{x})}[-\log h_\theta(\mathbf{x})[K+1]]$ encourages the module to reserve a certain degree of confidence for virtual class $\mathbf{p}_f$, thus measuring the out-of-distribution uncertainty for given samples and mitigating overconfident predictions. As shown in Fig. 2, $\mathbb{E}_{\mathbb{P}'_{\bar{\theta}}(\mathbf{x})}[-\log h_\theta(\mathbf{x})[K+1]]$ introduces an adversarial learning process, where we iteratively generate samples believed to have lower energies from $\mathbb{P}'_\theta(\mathbf{x})$ and then update the energy manifold to increase the energy of generated samples and decrease that of real samples. This process

effectively enhances the modeling of the known input distribution, making in-distribution samples have low energy and out-of-distribution data have high energy. Therefore, for a sample from unseen distributions, the module will produce predictions with large uncertainty ($h_\theta(\mathbf{x})[K+1]$) and low confidence due to the fact that the confidence must be lower than $1 - h_\theta(\mathbf{x})[K+1]$. Then, modules created in the future to handle these unknown distributions will dominate the final prediction. Hence, we achieve the *forward compatibility* for the future unseen categories.

### 3.3 ENERGY-BASED MODEL FUSION

After training the new module, we aim to fuse it with the prior ones to form a unifying classifier for all seen categories. Assuming that we have trained a unifying model $h_{\theta_o}$ for all the old tasks and $\sigma_o$ maps the label to the output index of $h_{\theta_o}$, a vanilla approach to combining the $h_{\theta_o}$ and $h_\theta$ is to redefine the energy function as

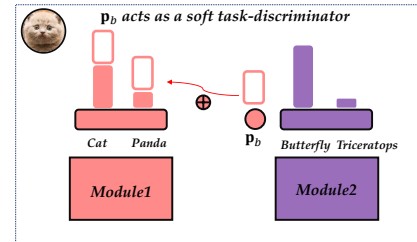

$$E_{\{\theta_o, \theta\}}(\mathbf{x}, y) = \begin{cases} -\log h_{\theta_o}(\mathbf{x})[\sigma_o(y)], & y \in \mathcal{Y}_o \\ -\log h_\theta(\mathbf{x})[\sigma(y)], & y \in \mathcal{Y}_n \end{cases}. \quad (8)$$

Then we have

Figure 3: $\mathbf{p}_b$ acts as a soft task-discriminator. It alleviates the task-bias and determines the dominant module for the ultimate prediction.

$$\mathbb{P}_{\{\theta, \theta_o\}}(y|\mathbf{x}) = \begin{cases} \frac{h_{\theta_o}(\mathbf{x})[\sigma_o(y)]}{\sum_{m=1}^{M} h_{\theta_o}(\mathbf{x})[m] + \sum_{k=1}^{K} h_\theta(\mathbf{x})[k]}, & y \in \mathcal{Y}_o \\ \frac{h_\theta(\mathbf{x})[\sigma(y)]}{\sum_{m=1}^{M} h_{\theta_o}(\mathbf{x})[m] + \sum_{k=1}^{K} h_\theta(\mathbf{x})[k]}, & y \in \mathcal{Y}_n \end{cases}. \quad (9)$$

However, this might cause task bias. Different modules may produce predictions with different entropies, the combined model has a tendency to modules with larger entropies. As shown in Fig. 3, Simply combing the modules as Eq. 9 leads to misclassification due to the larger entropy in module2. Considering that $\mathbf{p}_b$ measures the confidence for old categories, we redefine $E_{\{\theta_o, \theta\}}$ as

$$\begin{cases} -\log\{h_{\theta_o}(\mathbf{x})[\sigma_o(y)] + \alpha h_\theta(\mathbf{x})[0] + \beta\}, & y \in \mathcal{Y}_o \\ -\log h_\theta(\mathbf{x})[\sigma(y)], & y \in \mathcal{Y}_n \end{cases}. \quad (10)$$

Then we have

$$\mathbb{P}_{\{\theta, \theta_o\}}(y|\mathbf{x}) = \begin{cases} \frac{h_{\theta_o}(\mathbf{x})[\sigma(y)] + \alpha h_\theta(\mathbf{x})[0] + \beta}{\sum_{m=1}^{M}[h_{\theta_o}(\mathbf{x})[m] + \alpha h_\theta(\mathbf{x})[0] + \beta] + \sum_{k=1}^{K} h_\theta(\mathbf{x})[k]}, & y \in \mathcal{Y}_o \\ \frac{h_\theta(\mathbf{x})[\sigma(y)]}{\sum_{m=1}^{M}[h_{\theta_o}(\mathbf{x})[m] + \alpha h_\theta(\mathbf{x})[0] + \beta] + \sum_{k=1}^{K} h_\theta(\mathbf{x})[k]}, & y \in \mathcal{Y}_n \end{cases}. \quad (11)$$

We finetune $\alpha, \beta$ to minimize the negative log-likelihood on a tiny sub-dataset (exemplar-set), thus mitigating the task bias, namely

$$\alpha^*, \beta^* = \arg\min_{\alpha, \beta} \mathbb{E}_{\mathbb{P}_{real}(\mathbf{x}, y)}\left[-\log \mathbb{P}_{\{\theta, \theta_o\}}(y|\mathbf{x})\right]. \quad (12)$$

### 3.4 SUMMARY OF BEEF

To conclude, we propose a two-stage training approach: expansion and fusion. The expansion phase is intrinsically similar to naive fine-tuning, while trough our novel energy definition and gradient-equivalent optimization simplification, we expand the original $K$ classification model into a $K+2$ classification model through two additional prototypes (*e.g.* backward prototype $\mathbf{p}_b$ and forward prototype $\mathbf{p}_f$) to achieve the bi-directional compatibility. Specifically, $\mathbf{p}_b$ learns to measure the confidence on old categories by acting as the cluster prototype for all old samples and thus achieves the backward compatibility. In contrast, $\mathbf{p}_f$ learns to measure the open world uncertainty by acting as the cluster prototype for all samples generated by the built-in energy-based model to achieve the forward compatibility. By means of our energy-based framework, our model is able to learn the discrimination ability for the current task while synchronously modeling the input distribution. At the fusion phase, the confidence of the old prototype is added to the outputs of all old modules after passing through a learnable affine transformation, forming the ultimate prediction on all categories. The fusion strategy alleviates the possible task bias and effectively improves the performance compared with the naive fusion strategy in Eq. 9. Equipped with an energy-based nature, BEEF allows for a pleasant by-product: *test-time alignment*. That is, we apply SGLD (Welling & Teh, 2011) to decrease energy of given test samples. Through *test-time alignment*, some useful

characteristics shared by training samples are transferred into the test sample to reduce its energy, making it more explicit and producing more convincing predictions. We find that this process does improve the performance in many protocols though requiring additional computational resources in evaluation. One inevitable drawback of dynamic-based methods is that they suffer from a linearly growing number of parameters as the number of tasks, which might violate the memory usage limitations in CIL. Therefore, for practical usage and fair comparison, we apply the compression strategy following FOSTER (Wang et al., 2022a) to compress the expanded dual branch model into a single skeleton after each incremental learning session, which we call *BEEF-Compress*.

## 4 EMPIRICAL STUDIES

### 4.1 EXPERIMENTAL SETTINGS

**Datasets.** We validate our methods on widely used benchmarks of class-incremental learning CIFAR-100 (Krizhevsky et al., 2009) and ImageNet100/1000 (Deng et al., 2009). **CIFAR-100**: CIFAR-100 consists of 50,000 training images with 500 images per class, and 10,000 test images with 100 images per class. **ImageNet-1000**: ImageNet-1000 is a large scale dataset composed of about 1.28 million images for training and 50,000 for validation with 500 images per class. **ImageNet-100**: ImageNet-100 is composed of 100 classes randomly chosen from the original ImageNet-1000 dataset.

**CIAFR-100 Protocol.** For benchmark CIFAR-100, we evaluate two widely recognized protocols: **CIFAR-100 B0**: All the 100 classes are averagely divided into 5, 10, and 20 groups, respectively, *i.e.*, we should train all the 100 classes gradually with 20, 10, 5 classes per incremental session. In addition, models are allowed to save an exemplar-set to store no more than 2000 exemplars throughout all sessions. **CIFAR-100 B50**: We first train half of 100 classes at the base learning stage. Then, the rest 50 classes are averagely divided into 5, 10, and 25 groups, respectively, *i.e.*, we should train the rest 50 classes gradually with 10, 5, and 2 classes per incremental session. Slightly different from the first protocol, models are allowed to store no more than 20 exemplars for each class. Therefore, after training all the 100 classes, there are also no more than 2,000 exemplars.

**ImageNet Protocol.** For benchmarking ImageNet-100, we evaluate the performance on two different incremental tasks. In the first task, we split the 100 classes averagely into 10 sequential incremental sessions, and up to 2,000 exemplars are allowed to be stored in the exemplar-set. In the second task, models are first trained on 50 base classes and then sequentially trained on the 10 classes at the following incremental sessions (*i.e.*, 5 incremental sessions totally). The same as the protocol CIFAR-100 B50, models are allowed to store no more than 20 exemplars for each class. For benchmark ImageNet-1000, we train all the 1000 classes with 100 classes per step (10 steps in total) with an exemplar-set storing no more than 20,000 exemplars.

**Compared methods.** Our method and all baselines are implemented with Pytorch (Paszke et al., 2017) in PyCIL (Zhou et al., 2021a). We compare BEEF to strong regularization-based methods: iCaRL (Rebuffi et al., 2017), BiC (Wu et al., 2019), WA (Zhao et al., 2020), PodNet (Douillard et al., 2020), and Coil (Zhou et al., 2021b). Beside, we compare to the dynamic-structure-based method RPSNet (Rajasegaran et al., 2019), DER (Yan et al., 2021), Dytox (Douillard et al., 2021), FOSTER (Wang et al., 2022a). Apart from the above methods, we also compare with RMM (Liu et al., 2021b), which adjusts the memory partition strategy for new and old data. Among the compared methods, Dytox applies stronger neural architecture (Convit (d'Ascoli et al., 2021)) and additional data augmentation; FOSTER (Wang et al., 2022a) additionally uses the AutoAugmentation (Cubuk et al., 2019) to enhance the sample efficiency and improve classification accuracy. RMM (Liu et al., 2021b) achieves a better memory management strategy. With the same training memory, RMM chooses some new samples to train and discards the rest to allow restoring more old exemplars. Note that all these are orthogonal to the method itself, and therefore we combine the augmentation and memory management strategy with BEEF for fair comparison with these methods, which we regard as BEEF-Compress.

### 4.2 RESULTS AND ANALYSIS

**Comparison with SOTAs. CIFAR-100**: Table 2 , Fig. 4, and Fig. 7 summarize the experimental results on CIFAR-100 benchmark. We can observe that BEEF/BEEF-Compress achieves state-of-the-art performance under both B0 and B50 protocols. And more evident performance improvements are achieved under B50 protocol. Specifically, BEEF improves average accuracy by 3.12, 3.62, 5.45 compared to DER (prior state-of-the-art dynamic-based method) under protocol B50 with 5, 10, 25

| Methods | ImageNet-100 10 steps | | | | ImageNet-1000 10 steps | | | | ImageNet-100 B50 5 steps | | | |
| | top-1 | | top-5 | | top-1 | | top-5 | | top-1 | | top-5 | |
| | Avg | Last | Avg | Last | Avg | Last | Avg | Last | Avg | Last | Avg | Last |
|---|---|---|---|---|---|---|---|---|---|---|---|---|
| Bound | - | 81.50 | - | 95.10 | 89.27 | - | 79.89 | - | 81.20 | 81.50 | - | 95.10 |
| Replay | 59.21 | 41.00 | 81.67 | 68.44 | - | - | - | - | 55.73 | 43.38 | 79.17 | 71.08 |
| iCaRL | 67.11 | 50.98 | 84.08 | 71.52 | 38.4 | 22.7 | 63.7 | 44.0 | 62.56 | 53.69 | 81.75 | 73.58 |
| BiC | 65.13 | 42.40 | 84.04 | 64.14 | - | - | 84.0 | 73.2 | 66.36 | 49.9 | 83.59 | 70.42 |
| WA | 68.60 | 55.04 | 89.53 | 80.32 | 65.67 | 55.60 | 86.60 | 81.10 | 65.81 | 56.64 | 84.97 | 79.36 |
| PodNet | 64.03 | 45.40 | 84.06 | 68.58 | - | - | - | - | 73.84 | 62.94 | 89.51 | 83.52 |
| DER | 77.08 | 66.84 | 92.49 | 88.64 | 66.87 | 58.83 | 88.01 | 81.59 | 77.57 | 71.10 | 93.37 | 91.3 |
| DyTox | 71.85 | 57.94 | 90.72 | 83.52 | 68.14 | 59.75 | 87.03 | 82.93 | - | - | - | - |
| RPSNet | - | - | 87.90 | 74.00 | - | - | - | - | - | - | - | - |
| RMM | - | - | - | - | - | - | - | - | 79.52 | - | - | - |
| FOSTER | 78.71 | 70.14 | - | - | 68.34 | 58.53 | - | - | 80.22 | 75.52 | - | - |
| BEEF | 77.62 | 68.78 | **93.66** | **89.32** | 67.09 | 58.67 | 86.21 | 81.73 | 77.27 | 70.98 | 93.71 | **91.76** |
| BEEF-Compress | **79.34** | **71.12** | 93.30 | 88.94 | - | - | - | - | **80.52** | **74.62** | **94.10** | 91.42 |

Table 1: Performance on ImageNet. We report both average and last accuracy of top-1 and top-5.

| Methods | CIFAR-100 B0 | | | | | | CIFAR-100 B50 | | | | | |
| | 5 steps | | 10 steps | | 20 steps | | 5 steps | | 10 steps | | 25 steps | |
| | Avg | Last | Avg | Last | Avg | Last | Avg | Last | Avg | Last | Avg | Last |
|---|---|---|---|---|---|---|---|---|---|---|---|---|
| Bound | 80.40 | - | 80.41 | - | 81.49 | - | 79.89 | - | 79.91 | - | 80.37 | - |
| Replay | 60.63 | 43.08 | 59.38 | 41.01 | 58.20 | 38.69 | 52.70 | 41.26 | 43.43 | 36.16 | 41.09 | 37.50 |
| iCaRL | 67.60 | 54.23 | 64.64 | 49.52 | 63.51 | 45.12 | 61.79 | 52.04 | 52.69 | 44.64 | 52.10 | 45.57 |
| BiC | 67.63 | 56.22 | 65.38 | 50.79 | 62.38 | 43.08 | 61.68 | 49.19 | 57.04 | 43.82 | 53.61 | 40.38 |
| WA | 69.11 | 57.97 | 67.15 | 52.30 | 64.65 | 48.46 | 64.65 | 55.85 | 53.87 | 46.72 | 52.51 | 44.90 |
| Coil | 68.26 | - | 65.48 | - | 62.98 | - | - | - | - | - | - | - |
| DER | 71.15 | 62.4 | 69.94 | 58.59 | 67.98 | 53.95 | 68.58 | 61.94 | 66.40 | 58.85 | 60.66 | 49.30 |
| RPSNet | 70.5 | - | 68.6 | - | - | - | - | - | - | - | - | - |
| DyTox | - | - | 71.50 | 57.76 | 68.86 | 51.47 | - | - | - | - | - | - |
| RMM | - | - | - | - | - | - | 68.86 | - | 67.61 | - | 66.21 | - |
| FOSTER | 72.54 | 64.55 | 72.81 | **62.54** | 70.65 | 56.28 | 70.10 | 64.01 | 67.94 | 60.44 | 63.83 | 54.31 |
| BEEF | 72.31 | **62.58** | 71.94 | 60.98 | 69.84 | 56.71 | **71.70** | **65.24** | 70.71 | **63.51** | **66.11** | **54.36** |
| BEEF-Compress | **73.05** | 62.48 | **72.93** | 61.45 | **71.69** | **57.06** | 71.58 | 64.54 | **71.70** | 61.19 | 64.32 | **54.81** |

Table 2: Performance on CIFAR-100. We report both the top-1 average and last accuracy.

incremental sessions. **ImageNet-100/1000**: Tabel 1 and Fig. 8 summarize the experimental results on both ImageNet-100 and ImageNet-1000 benchmarks. We can observe that BEEF, especially BEEF-Compress achieves very competitive performance compared to prior methods. Specifically, BEEF improves average accuracy by 0.63, 0.3 under ImageNet-100 protocols. It is also worth noting that the performance improvement of BEEF on ImageNet is relatively less significant compared to CIFAR-100, which may be attributed to the larger and more complex nature of the ImageNet dataset, making it more challenging to find suitable cluster prototypes as forward/backward prototypes.

**Comparison under imbalanced exemplar-set.** To compare the performance of different methods with imbalanced exemplar-set, we propose three different exemplar selection strategies: *exp*, *random*, and *half-half*. Refer to Appendix C.1 for details. Fig. 4 displays the average accuracy changes of different methods after each incremental session on CIFAR-100 B50 with 5 steps. Fig. 5(c) illustrates the performance when exemplars are randomly sampled from all the available old instances. Though the exemplar-set is statistically balanced, prior methods encounters a performance drop and the gap between BEEF and prior methods is enlarged. Fig. 5(b) and Fig. 5(a) illustrate the performance changes under extreme class imbalance and sample from half classes missing, respectively. Although the performance of prior methods declines dramatically, BEEF maintains its effectiveness under these two imbalanced protocols. BEEF achieves more than 10% performance gain under these challenging settings. In addition, since some classes have no exemplars stored in the exemplar-set to calculate the class center, iCaRL based on NCM-classifier (Mensink et al., 2013) fails in the base training phase.

## 4.3 ABLATION STUDIES

**Ablations of key components in BEEF.** To verify the effectiveness of the components in BEEF, we conduct ablation studies on CIFAR-100 B50 with 5 incremental sessions. As shown in Table 3, the average and last accuracy gradually increase as we add more components. It is notable that after the fusion strategy, the unifying model get a large increase. Besides, by learning energy manifolds with the forward prototype $\mathbf{p}_f$ and energy alignment, we can further improve the performance.

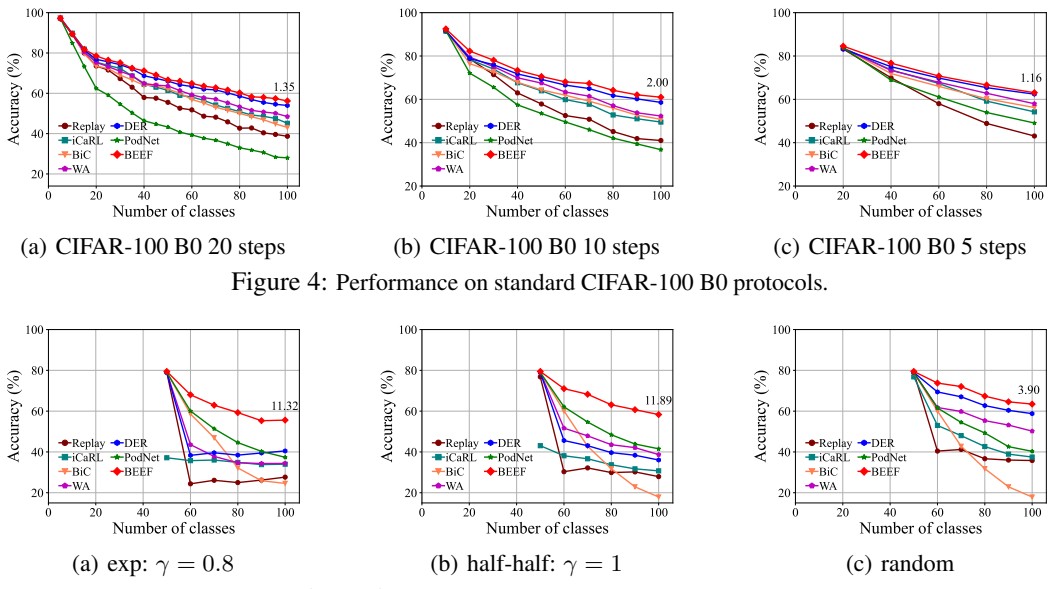

(a) CIFAR-100 B0 20 steps     (b) CIFAR-100 B0 10 steps     (c) CIFAR-100 B0 5 steps

Figure 4: Performance on standard CIFAR-100 B0 protocols.

(a) exp: $\gamma = 0.8$     (b) half-half: $\gamma = 1$     (c) random

Figure 5: Performance on imbalanced protocols.

| Expansion | | Fusion | | Avg | Last |
|---|---|---|---|---|---|
| backward compatible | forward compatible | task discriminator | energy alignment | | |
| ✓ | | | | 63.60 | 55.16 |
| ✓ | ✓ | | | 64.50 | 56.50 |
| ✓ | | ✓ | | 70.51 | 64.43 |
| ✓ | ✓ | ✓ | | 70.92 | 64.79 |
| ✓ | ✓ | ✓ | ✓ | 71.75 | 65.24 |

Table 3: Ablations of key components in BEEF. We report the average and last accuracy on CIFAR-100 B50 with 5 incremental sessions.

**Sensitive study of hyper-parameters.** There are three hyper-parameters in BEEF when modeling the energy manifold: hidden layer where energy modeling, the trade-off coefficient $\lambda$, and the number of forward prototypes $F$ in Stable-BEEF. As shown in Fig. 6(a) we conduct experiments on three different hidden layers for modeling the energy manifold, and the experimental results show BEEF the robustness to the choice of different hidden layer. We also change the trade-off coefficients $\lambda$ from $\{0, 1e-1, 1e-2, 1e-3\}$ and the number of forward prototypes $F$ from $\{1, 5, 20, 50\}$, and the average accuracies on CIFAR-100 B50 with 5 incremental sessions are shown in Fig. 6(b). We can see that with the increase of $F$ and $\lambda$, the accuracy has an upward trend.

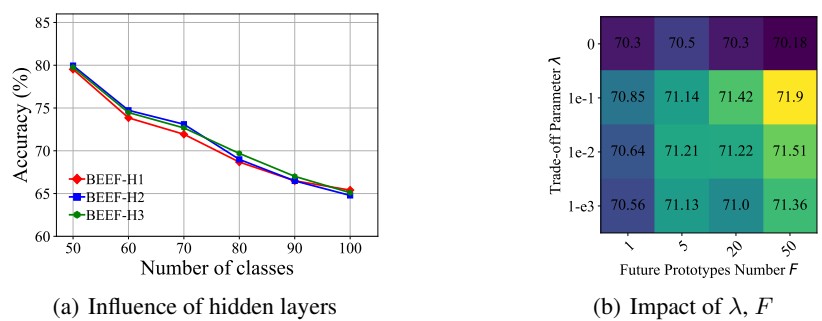

(a) Influence of hidden layers     (b) Impact of $\lambda$, $F$

Figure 6: Sensitive studies of hyper-parameters.

CONCLUSION. In this work we presented BEEF for achieving efficient class-incremental learning. Under this framework, we efficiently train a specific module for the current task while achieving bi-directional compatibility and fuse it with the prior model under minimal effort. BEEF is equipped with a theoretical analysis showing that its training process is inherently the modeling of an energy-based model. Compression strategies can be applied to address the issue of growing storage overhead.

ACKNOWLEDGMENT. This work is partially supported by NSFC (61921006, 62006112, 62250069), NSF of Jiangsu Province (BK20200313), Collaborative Innovation Center of Novel Software Technology and Industrialization.

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

# Appendix of BEEF

CONTENTS

## A   PROOF FOR THM. 3.1 AND THM. 3.2

Here we provide the detailed proofs for Thm. 3.1 and Thm. 3.2.

**Theorem 3.1** (Marginal Distribution Maximum Likelihood Estimation). *Defining $E'_\theta(\mathbf{x}) = -\log h_\theta(\mathbf{x})[K + 1]$ and its corresponding marginal distribution as $\mathbb{P}'_\theta(\mathbf{x})$, the optimization of $\mathbb{E}_{\mathbb{P}_{real}(\mathbf{x})}[-\log \mathbb{P}_\theta(\mathbf{x})]$ is equivalent to that of $\mathbb{E}_{\mathbb{P}_{real}(\mathbf{x})}\left[-\log \sum_{k=0}^{K} h_\theta(\mathbf{x})[k]\right] + \lambda_{\bar{\theta}}\mathbb{E}_{\mathbb{P}'_{\bar{\theta}}(\mathbf{x})}[-\log h_\theta(\mathbf{x})[K + 1]]$ when gradient descend is applied, where $\lambda_{\bar{\theta}}$ is the ratio of the normalizing constants determined by $E'_\theta(\mathbf{x})$ and $E_\theta(\mathbf{x})$, and $\bar{\theta}$ means that parameters of $\theta$ is frozen (i.e., instances sampled from $\mathbb{P}'_{\bar{\theta}}(\mathbf{x})$ are detached).*

*Proof.* Since $\mathbb{P}_\theta(\mathbf{x}) = \frac{\exp(-E_\theta(\mathbf{x}))}{\sum_{\mathbf{x}'} \exp(-E_\theta(\mathbf{x}))}$, we have

$$\mathbb{E}_{\mathbb{P}_{real}(\mathbf{x})}[-\log \mathbb{P}_\theta(\mathbf{x})] = \mathbb{E}_{\mathbb{P}_{real}(\mathbf{x})}[E_\theta(\mathbf{x})] + \log \sum_{\mathbf{x}'} \exp(-E_\theta(\mathbf{x}')). \tag{13}$$

Take the gradient of the right part in Eq. 13, and we have

$$
\begin{aligned}
& \nabla_\theta \log \sum_{\mathbf{x}'} \exp(-E_\theta(\mathbf{x}')) \\
=& \sum_{\mathbf{x}} \frac{\exp(-E_\theta(\mathbf{x}))}{\sum_{\mathbf{x}'} \exp(-E_\theta(\mathbf{x}'))} \nabla_\theta E_\theta(\mathbf{x}) \\
=& \sum_{\mathbf{x}} \frac{\sum_{k=0}^{K} h_\theta(\mathbf{x})[k]}{\sum_{\mathbf{x}'} \exp(-E_\theta(\mathbf{x}'))} \cdot \frac{\nabla_\theta \sum_{k=0}^{K} h_\theta(\mathbf{x})[k]}{\sum_{k=0}^{K} h_\theta(\mathbf{x})[k]} \\
=& \sum_{\mathbf{x}} \frac{\nabla_\theta (1 - h_\theta(\mathbf{x})[K+1])}{\sum_{\mathbf{x}'} \exp(-E_\theta(\mathbf{x}'))} \\
=& -\sum_{\mathbf{x}} \frac{\nabla_\theta h_\theta(\mathbf{x})[K+1]}{\sum_{\mathbf{x}'} \exp(-E_\theta(\mathbf{x}'))} \\
=& -\sum_{\mathbf{x}} \frac{\sum_{\mathbf{x}'} \exp(-E'_\theta(\mathbf{x}'))}{\sum_{\mathbf{x}'} \exp(-E_\theta(\mathbf{x}'))} \cdot \frac{\exp(-E'_\theta(\mathbf{x}))}{\sum_{\mathbf{x}'} \exp(-E'_\theta(\mathbf{x}'))} \cdot \frac{\nabla h_\theta(\mathbf{x})[K+1]}{\exp(-E'_\theta(\mathbf{x}))} \\
=& -\sum_{\mathbf{x}} \frac{\sum_{\mathbf{x}'} \exp(-E'_\theta(\mathbf{x}'))}{\sum_{\mathbf{x}'} \exp(-E_\theta(\mathbf{x}'))} \cdot \frac{\exp(-E'_\theta(\mathbf{x}))}{\sum_{\mathbf{x}'} \exp(-E'_\theta(\mathbf{x}'))} \cdot \frac{\nabla h_\theta(\mathbf{x})[K+1]}{h_\theta(\mathbf{x})[K+1]} \\
=& -\sum_{\mathbf{x}} \frac{Z'_\theta}{Z_\theta} \cdot \frac{\exp(-E'_\theta(\mathbf{x}))}{\sum_{\mathbf{x}'} \exp(-E'_\theta(\mathbf{x}'))} \cdot \nabla_\theta \log h_\theta(\mathbf{x})[K+1] \\
=& -\frac{Z'_\theta}{Z_\theta} \mathbb{E}_{\mathbb{P}'_\theta(\mathbf{x})} \left[ \nabla_\theta \log h_\theta(\mathbf{x})[K+1] \right] \\
=& -\nabla_\theta \left\{ \frac{Z'_{\bar{\theta}}}{Z_{\bar{\theta}}} \mathbb{E}_{\mathbb{P}'_{\bar{\theta}}(\mathbf{x})} \left[ \log h_\theta(\mathbf{x})[K+1] \right] \right\}.
\end{aligned}
\tag{14}
$$

where $\bar{\theta}$ represents that $\theta$ is frozen (*i.e.*, gradients are not computed), $Z'_{\bar{\theta}} = \sum_{\mathbf{x}'} \exp(-E'_\theta(\mathbf{x}'))$ is the normalizing constant for energy $E'_\theta(\mathbf{x})$, and $Z_\theta$ is the normalizing constant for energy $E_\theta(\mathbf{x})$. Hence, the objective $\mathbb{E}_{\mathbb{P}_{real}(\mathbf{x})} \left[ -\log \mathbb{P}_\theta(\mathbf{x}) \right]$ is equivalent to

$$
\mathbb{E}_{\mathbb{P}_{real}(\mathbf{x})} \left[ -\log \sum_{k=0}^{K} h_\theta(\mathbf{x})[k] \right] + \lambda_{\bar{\theta}} \mathbb{E}_{\mathbb{P}_{\bar{\theta}}(\mathbf{x}')} \left[ -\log h_\theta(\mathbf{x})[K+1] \right],
\tag{15}
$$

where $\lambda_{\bar{\theta}} = \frac{Z'_{\bar{\theta}}}{Z_{\bar{\theta}}}$. □

**Theorem 3.2** (Conditional Distribution Maximum Likelihood Estimation). *With preliminaries from Thm. 3.1, the optimization of $\mathbb{E}_{\mathbb{P}_{real}(\mathbf{x},y)} \left[ -\log \mathbb{P}_\theta(y|\mathbf{x}) \right]$ is equivalent to that of $\mathbb{E}_{\mathbb{P}_{real}(\mathbf{x},y)} \left[ -\log h_\theta(\mathbf{x})[\sigma'(y)] \right] + \mu_{\bar{\theta}} \mathbb{E}_{\mathbb{P}_{real}(\mathbf{x})} \left[ -\log h_\theta(\mathbf{x})[K+1] \right]$ when gradient descend is applied, where $\mu_{\bar{\theta}} = \frac{h_{\bar{\theta}}(\mathbf{x})[K+1]}{\sum_{k=0}^{K} h_{\bar{\theta}}(\mathbf{x})[k]}$, $\sigma'(y) = \begin{cases} \sigma(y), & y \in \mathcal{Y}_n \\ 0, & y \in \mathcal{Y}_o \end{cases}$.*

*Proof.* Considering the definition of $\mathbb{P}_\theta(y|\mathbf{x})$ in Eq. 2, we have

$$
\begin{aligned}
& \mathbb{E}_{\mathbb{P}_{real}(\mathbf{x},y)} \left[ -\log \mathbb{P}_\theta(y \mid \mathbf{x}) \right] \\
=& \mathbb{E}_{\mathbb{P}_{real}(\mathbf{x},y)} \left[ E_\theta(\mathbf{x},y) + \log \sum_{y' \in \mathcal{Y}_n \cup \mathcal{Y}_o} \exp(-E_\theta(\mathbf{x},y')) \right] \\
=& \mathbb{E}_{\mathbb{P}_{real}(\mathbf{x},y) \cap y \in \mathcal{Y}_o} \left[ -\log \frac{h_\theta(\mathbf{x})[0]}{M} \right] + \mathbb{E}_{\mathbb{P}_{real}(\mathbf{x},y) \cap y \in \mathcal{Y}_n} \left[ -\log h_\theta(\mathbf{x})[\sigma(y)] \right] + \\
& \mathbb{E}_{\mathbb{P}_{real}(\mathbf{x})} \left[ \log \sum_{k=0}^{K} h_\theta(\mathbf{x})[k] \right]
\end{aligned}
\tag{16}
$$

The gradient of $\mathbb{E}_{\mathbb{P}_{real}(\mathbf{x},y) \cap y \in \mathcal{Y}_o} \left[ -\log \frac{h_\theta(\mathbf{x})[0]}{M} \right]$ with respect to $\theta$ is equal to that of $\mathbb{E}_{\mathbb{P}_{real}(\mathbf{x},y) \cap y \in \mathcal{Y}_o} \left[ -\log h_\theta(\mathbf{x})[0] \right]$. Therefore, define $\sigma'(y) = \begin{cases} \sigma(y), & y \in \mathcal{Y}_n \\ 0, & y \in \mathcal{Y}_o \end{cases}$, and the first two components in Eq. 16 can be written in a unifying form, that is

$$\mathbb{E}_{\mathbb{P}_{real}(\mathbf{x},y) \cap y \in \mathcal{Y}_o \cup \mathcal{Y}_n} \left[ -\log h_\theta(\mathbf{x})[\sigma'(y)] \right] . \tag{17}$$

For the last component in Eq. 16, we have

$$
\begin{aligned}
& \nabla_\theta \log \sum_{k=0}^{K} h_\theta(\mathbf{x})[k] \\
=& \frac{\nabla_\theta (1 - h_\theta(\mathbf{x})[K+1])}{\sum_{k=0}^{K} h_\theta(\mathbf{x})[k]} \\
=& -\frac{\nabla_\theta h_\theta(\mathbf{x})[K+1]}{\sum_{k=0}^{K} h_\theta(\mathbf{x})[k]} \\
=& -\frac{h_\theta(\mathbf{x})[K+1]}{\sum_{k=0}^{K} h_\theta(\mathbf{x})[k]} \cdot \frac{\nabla_\theta h_\theta(\mathbf{x})[K+1]}{h_\theta(\mathbf{x})[K+1]} \\
=& -\frac{h_\theta(\mathbf{x})[K+1]}{\sum_{k=0}^{K} h_\theta(\mathbf{x})[k]} \cdot \nabla_\theta \log h_\theta(\mathbf{x})[K+1] \\
=& \nabla_\theta \left\{ -\frac{h_{\bar{\theta}}(\mathbf{x})[K+1]}{\sum_{k=0}^{K} h_{\bar{\theta}}(\mathbf{x})[k]} \cdot \log h_\theta(\mathbf{x})[K+1] \right\}
\end{aligned}
\tag{18}
$$

Therefore, the Objective $\mathbb{E}_{\mathbb{P}_{real}(\mathbf{x},y)} \left[ -\log \mathbb{P}_\theta(y \mid \mathbf{x}) \right]$ is also equivalent to

$$\mathbb{E}_{\mathbb{P}_{real}(\mathbf{x},y)} \left[ -\log h_\theta(\mathbf{x})[\sigma'(y)] \right] + \mu_{\bar{\theta}} \mathbb{E}_{\mathbb{P}_{real}(\mathbf{x})} \left[ -\log h_\theta(\mathbf{x})[K+1] \right] , \tag{19}$$

where $\mu_{\bar{\theta}} = \frac{h_{\bar{\theta}}(\mathbf{x})[K+1]}{\sum_{k=0}^{K} h_{\bar{\theta}}(\mathbf{x})[k]}$. $\qquad \square$

## B  PROOFS FOR STABLE-BEEF

As we have claimed in Sec. 3 and Sec. 4, we expand the original BEEF to have multiple backward and forward prototypes (dubbed as Stable-BEEF), which stabilizes the training process and improves the performance. Here we give a formal illustration of the expandable form and provide proof of it.

First, at the $t^{th}$ incremental session, instead of creating one backward prototype $\mathbf{p}_b$ to measure the confidence for all the old tasks, we create a backward prototype for each old task. This expansion prompts the new module to learn a discriminator for each old task, thereby improving the performance and stability of training especially when there are clear domain shifts among old tasks. Therefore, there are $t-1$ backward prototypes for measuring the confidence of old tasks. Furthermore, in Stable-BEEF, we create multiple forward prototypes during the training. Concretely, for samples generated through the marginal distribution defined by the energy function, we assign the most-likely pseudo labels to them. We set the number of forward prototypes to a constant number $F$. After introducing these multiple prototypes, i.e., $\mathbf{p}_b$s and $\mathbf{p}_f$s, we should redefine $h_\theta : \mathcal{X} \longrightarrow \Delta^{K+F+(t-2)}$.

Given an input-label pair $(\mathbf{x}, y) \in \cup_{i=1}^{t} \mathcal{X}_i \times \cup_{i=1}^{t} \mathcal{Y}_i$, we define the energy $E_\theta(\mathbf{x}, y)$ as

$$E_\theta(\mathbf{x}, y) = \begin{cases} -\log h_\theta(\mathbf{x})[\sigma(y)], & y \in \mathcal{Y}_t \\ -\log \left( h_\theta(\mathbf{x})[\sigma(y)] / |\mathcal{Y}_i| \right), & y \in \mathcal{Y}_i, \quad i = 1, 2, \ldots, t-1 \end{cases}, \tag{20}$$

where $\sigma : \cup_{i=1}^{t} \mathcal{Y}_i \longrightarrow K + F + (t-1)$ maps new labels to their $K$ class-corresponding prototypes and maps old labels to their $t-1$ task-corresponding backward prototypes.

Therefore, similar to Eq. 2, the conditional probability density and marginal probability density for Stable-BEEF can be re-formulated as :

$$\mathbb{P}_\theta(y|\mathbf{x}) = \begin{cases} \frac{h_\theta(\mathbf{x})[\sigma(y)]}{\sum_{k=0}^{K+t-2} h_\theta(\mathbf{x})[k]}, & y \in \mathcal{Y}_t \\ \frac{h_\theta(\mathbf{x})[\sigma(y)]}{|\mathcal{Y}_i| \sum_{k=0}^{K+t-2} h_\theta(\mathbf{x})[k]}, & y \in \mathcal{Y}_i, i = 1, \ldots, t-1 \end{cases}, \quad \mathbb{P}_\theta(\mathbf{x}) = \frac{\sum_{k=0}^{K+t-2} h_\theta(\mathbf{x})[k]}{\sum_{\mathbf{x}'} \sum_{k=0}^{K+t-2} h_\theta(\mathbf{x}')[k]} . \tag{21}$$

And then we can induce that the energy function in $\mathbf{P}_\theta(\mathbf{x})$ is formulated as

$$E_\theta(\mathbf{x}) = -\log \sum_{k=0}^{K+t-2} h_\theta(\mathbf{x})[k] . \tag{22}$$

Like Eq. 4, Stable-BEEF also estimates the joint distribution $\mathbb{P}_\theta(\mathbf{x}, y)$, that is

$$\arg\min_\theta \quad \mathbb{E}_{\mathbb{P}_{real}(\mathbf{x})} \left[ -\log \mathbb{P}_\theta(\mathbf{x}) \right] + \mathbb{E}_{\mathbb{P}_{real}(\mathbf{x},y)} \left[ -\log \mathbb{P}_\theta(y|\mathbf{x}) \right] . \tag{23}$$

And same to the original BEEF, Stable-BEEF also aims to find its gradient equivalent optimization objective to avoid the intractable normalizing constant in the joint distribution defined by the energy $\mathbf{E}_\theta(\mathbf{x}, y)$.

**Theorem B.1** (Marginal Distribution Maximum Likelihood Estimation for Stable-BEEF). *Defining* $E'_\theta(\mathbf{x}) = -\log \sum_{k=K+t-1}^{K+t-2+F} h_\theta(\mathbf{x})[k]$ *and its corresponding marginal distribution as* $\mathbb{P}'_\theta(\mathbf{x})$, *the optimization of* $\mathbb{E}_{\mathbb{P}_{real}(\mathbf{x})} \left[ -\log \mathbb{P}_\theta(\mathbf{x}) \right]$ *is equivalent to that of* $\mathbb{E}_{\mathbb{P}_{real}(\mathbf{x})} \left[ -\log \sum_{k=0}^{K+t-2} h_\theta(\mathbf{x})[k] \right] + \lambda_{\bar{\theta}} \mathbb{E}_{\mathbb{P}'_{\bar{\theta}}(\mathbf{x})} \left[ -\log \sum_{k=K+t-1}^{K+t-2+F} h_\theta(\mathbf{x})[k] \right]$ *when gradient descend is applied, where* $\lambda_{\bar{\theta}}$ *is the ratio of the normalizing constants determined by* $E'_\theta(\mathbf{x})$ *and* $E_\theta(\mathbf{x})$, *and* $\bar{\theta}$ *means that parameters of* $\theta$ *is frozen (i.e., instances sampled from* $\mathbb{P}'_{\bar{\theta}}(\mathbf{x})$ *are detached).*

*Proof.* Since $\mathbb{P}_\theta(\mathbf{x}) = \frac{\exp(-E_\theta(\mathbf{x}))}{\sum_{\mathbf{x}'} \exp(-E_\theta(\mathbf{x}))}$, we have

$$\mathbb{E}_{\mathbb{P}_{real}(\mathbf{x})} \left[ -\log \mathbb{P}_\theta(\mathbf{x}) \right] = \mathbb{E}_{\mathbb{P}_{real}(\mathbf{x})} \left[ E_\theta(\mathbf{x}) \right] + \log \sum_{\mathbf{x}'} \exp(-E_\theta(\mathbf{x}')) . \tag{24}$$

Take the gradient of the right part in Eq. 24, and we have

$$
\begin{aligned}
&\nabla_\theta \log \sum_{\mathbf{x}'} \exp(-E_\theta(\mathbf{x}')) \\
&= \sum_{\mathbf{x}} \frac{\exp(-E_\theta(\mathbf{x}))}{\sum_{\mathbf{x}'} \exp(-E_\theta(\mathbf{x}'))} \nabla_\theta E_\theta(\mathbf{x}) \\
&= \sum_{\mathbf{x}} \frac{\sum_{k=0}^{K+t-2} h_\theta(\mathbf{x})[k]}{\sum_{\mathbf{x}'} \exp(-E_\theta(\mathbf{x}'))} \cdot \frac{\nabla_\theta \sum_{k=0}^{K+t-2} h_\theta(\mathbf{x})[k]}{\sum_{k=0}^{K+t-2} h_\theta(\mathbf{x})[k]} \\
&= \sum_{\mathbf{x}} \frac{\nabla_\theta (1 - \sum_{k=K+t-1}^{K+t-2+F} h_\theta(\mathbf{x})[k])}{\sum_{\mathbf{x}'} \exp(-E_\theta(\mathbf{x}'))} \\
&= -\sum_{\mathbf{x}} \frac{\nabla_\theta \sum_{k=K+t-1}^{K+t-2+F} h_\theta(\mathbf{x})[k]}{\sum_{\mathbf{x}'} \exp(-E_\theta(\mathbf{x}'))} \\
&= -\sum_{\mathbf{x}} \frac{\sum_{\mathbf{x}'} \exp(-E'_\theta(\mathbf{x}'))}{\sum_{\mathbf{x}'} \exp(-E_\theta(\mathbf{x}'))} \cdot \frac{\exp(-E'_\theta(\mathbf{x}))}{\sum_{\mathbf{x}'} \exp(-E'_\theta(\mathbf{x}'))} \cdot \frac{\nabla \sum_{k=K+t-1}^{K+t-2+F} h_\theta(\mathbf{x})[k]}{\exp(-E'_\theta(\mathbf{x}))} \quad \tag{25} \\
&= -\sum_{\mathbf{x}} \frac{\sum_{\mathbf{x}'} \exp(-E'_\theta(\mathbf{x}'))}{\sum_{\mathbf{x}'} \exp(-E_\theta(\mathbf{x}'))} \cdot \frac{\exp(-E'_\theta(\mathbf{x}))}{\sum_{\mathbf{x}'} \exp(-E'_\theta(\mathbf{x}'))} \cdot \frac{\nabla \sum_{k=K+t-1}^{K+t-2+F} h_\theta(\mathbf{x})[k]}{\sum_{k=K+t-1}^{K+t-2+F} h_\theta(\mathbf{x})[k]} \\
&= -\sum_{\mathbf{x}} \frac{Z'_\theta}{Z_\theta} \cdot \frac{\exp(-E'_\theta(\mathbf{x}))}{\sum_{\mathbf{x}'} \exp(-E'_\theta(\mathbf{x}'))} \cdot \nabla_\theta \log \sum_{k=K+t-1}^{K+t-2+F} h_\theta(\mathbf{x})[k] \\
&= -\frac{Z'_\theta}{Z_\theta} \mathbb{E}_{\mathbb{P}'_\theta(\mathbf{x})} \left[ \nabla_\theta \log \sum_{k=K+t-1}^{K+t-2+F} h_\theta(\mathbf{x})[k] \right] \\
&= -\nabla_\theta \left\{ \frac{Z'_{\bar{\theta}}}{Z_{\bar{\theta}}} \mathbb{E}_{\mathbb{P}'_{\bar{\theta}}(\mathbf{x})} \left[ \log \sum_{k=K+t-1}^{K+t-2+F} h_\theta(\mathbf{x})[k] \right] \right\} .
\end{aligned}
$$

where $\bar{\theta}$ represents that $\theta$ is frozen (*i.e.*, gradients are not computed), $Z'_{\bar{\theta}} = \sum_{\mathbf{x}'} \exp(-E'_\theta(\mathbf{x}'))$ is the normalizing constant for energy $E'_\theta(\mathbf{x})$, and $Z_\theta$ is the normalizing constant for energy $E_\theta(\mathbf{x})$. Hence, the objective $\mathbb{E}_{\mathbb{P}_{real}(\mathbf{x})} [- \log \mathbb{P}_\theta(\mathbf{x})]$ is equivalent to

$$\mathbb{E}_{\mathbb{P}_{real}(\mathbf{x})} \left[ - \log \sum_{k=0}^{K+t-2} h_\theta(\mathbf{x})[k] \right] + \lambda_{\bar{\theta}} \mathbb{E}_{\mathbb{P}_{\bar{\theta}}(\mathbf{x}')} \left[ - \log \sum_{k=K+t-1}^{K+t-2+F} h_\theta(\mathbf{x})[K+1] \right] , \quad (26)$$

where $\lambda_{\bar{\theta}} = \frac{Z'_{\bar{\theta}}}{Z_{\bar{\theta}}}$. $\qquad\qquad\qquad\qquad\qquad\qquad\qquad\qquad\qquad\qquad\qquad\qquad\qquad\qquad$ □

**Theorem B.2** (Conditional Distribution Maximum Likelihood Estimation for Stable-BEEF). *With preliminaries from Thm. B.1, the optimization of $\mathbb{E}_{\mathbb{P}_{real}(\mathbf{x},y)} [- \log \mathbb{P}_\theta(y|\mathbf{x})]$ is equivalent to that of $\mathbb{E}_{\mathbb{P}_{real}(\mathbf{x},y)} [- \log h_\theta(\mathbf{x})[\sigma(y)]] + \mu_{\bar{\theta}} \mathbb{E}_{\mathbb{P}_{real}(\mathbf{x})} \left[ - \log \sum_{k=K+t-1}^{K+t-2+F} h_\theta(\mathbf{x})[k] \right]$ when gradient descend is applied, where $\mu_{\bar{\theta}} = \frac{\sum_{k=K+t-1}^{K+t-2+F} h_{\bar{\theta}}(\mathbf{x})[K+1]}{\sum_{k=0}^{K+t-2} h_{\bar{\theta}}(\mathbf{x})[k]}$.*

*Proof.* Considering the definition of $\mathbb{P}_\theta(y|\mathbf{x})$ in Eq. 2, we have

$$\mathbb{E}_{\mathbb{P}_{real}(\mathbf{x},y)} [- \log \mathbb{P}_\theta(y \mid \mathbf{x})]$$

$$= \mathbb{E}_{\mathbb{P}_{real}(\mathbf{x},y)} \left[ E_\theta(\mathbf{x},y) + \log \sum_{y' \in \cup_{i=1}^t \mathcal{Y}_i} \exp(-E_\theta(\mathbf{x},y')) \right]$$

$$= \sum_{i=1}^{t-1} \mathbb{E}_{\mathbb{P}_{real}(\mathbf{x},y) \cap y \in \mathcal{Y}_i} \left[ - \log \frac{h_\theta(\mathbf{x})[\sigma(y)]}{|\mathcal{Y}_i|} \right] + \mathbb{E}_{\mathbb{P}_{real}(\mathbf{x},y) \cap y \in \mathcal{Y}_t} [- \log h_\theta(\mathbf{x})[\sigma(y)]] + \quad (27)$$

$$\mathbb{E}_{\mathbb{P}_{real}(\mathbf{x})} \left[ \log \sum_{k=0}^{K+t-2} h_\theta(\mathbf{x})[k] \right]$$

The gradient of $\mathbb{E}_{\mathbb{P}_{real}(\mathbf{x},y) \cap y \in \mathcal{Y}_i} \left[ - \log \frac{h_\theta(\mathbf{x})[\sigma(y)]}{|\mathcal{Y}_i|} \right]$ with respect to $\theta$ is equal to that of $\mathbb{E}_{\mathbb{P}_{real}(\mathbf{x},y) \cap y \in \mathcal{Y}_i} [- \log h_\theta(\mathbf{x})[\sigma(y)]]$ ($i = 1, 2, \ldots, t-1$) no matter whatever $|\mathcal{Y}_i|$ is. Therefore, the first $t$ components in Eq. 27 can be written in a unifying form, that is

$$\mathbb{E}_{\mathbb{P}_{real}(\mathbf{x},y) \cap y \in \cup_{i=1}^t \mathcal{Y}_i} [- \log h_\theta(\mathbf{x})[\sigma(y)]] , \quad (28)$$

where For the last component in Eq. 16, we have

$$\nabla_\theta \log \sum_{k=0}^{K+t-2} h_\theta(\mathbf{x})[k]$$

$$= \frac{\nabla_\theta (1 - \sum_{k=K+t-1}^{K+t-2+F} h_\theta(\mathbf{x})[k])}{\sum_{k=0}^{K+t-2} h_\theta(\mathbf{x})[k]}$$

$$= - \frac{\nabla_\theta \sum_{k=K+t-1}^{K+t-2+F} h_\theta(\mathbf{x})[k]}{\sum_{k=0}^{K+t-2} h_\theta(\mathbf{x})[k]} \quad (29)$$

$$= - \frac{\sum_{k=K+t-1}^{K+t-2+F} h_\theta(\mathbf{x})[k]}{\sum_{k=0}^{K+t-2} h_\theta(\mathbf{x})[k]} \cdot \frac{\nabla_\theta \sum_{k=K+t-1}^{K+t-2+F} h_\theta(\mathbf{x})[k]}{\sum_{k=K+t-1}^{K+t-2+F} h_\theta(\mathbf{x})[k]}$$

$$= - \frac{\sum_{k=K+t-1}^{K+t-2+F} h_\theta(\mathbf{x})[k]}{\sum_{k=0}^{K+t-2} h_\theta(\mathbf{x})[k]} \cdot \nabla_\theta \log h_\theta(\mathbf{x})[K+1]$$

$$= \nabla_\theta \left\{ - \frac{\sum_{k=K+t-1}^{K+t-2+F} h_\theta(\mathbf{x})[k]}{\sum_{k=0}^{K+t-2} h_\theta(\mathbf{x})[k]} \cdot \log \sum_{k=K+t-1}^{K+t-2+F} h_\theta(\mathbf{x})[k] \right\}$$

Therefore, the Objective $\mathbb{E}_{\mathbb{P}_{real}(\mathbf{x},y)} [- \log \mathbb{P}_\theta(y \mid \mathbf{x})]$ is also equivalent to

$$\mathbb{E}_{\mathbb{P}_{real}(\mathbf{x},y)} [- \log h_\theta(\mathbf{x})[\sigma(y)]] + \mu_{\bar{\theta}} \mathbb{E}_{\mathbb{P}_{real}(\mathbf{x})} \left[ - \log \sum_{k=K+t-1}^{K+t-2+F} h_\theta(\mathbf{x})[k] \right] , \quad (30)$$

where $\mu_{\bar{\theta}} = \frac{\sum_{k=K+t-1}^{K+t-2+F} h_\theta(\mathbf{x})[k]}{\sum_{k=0}^{K+t-2} h_\theta(\mathbf{x})[k]}$. $\qquad\qquad\qquad\qquad\qquad\qquad\qquad\qquad\qquad\square$

Combining Thm. 26 and Thm. 30, we get the final optimization objective

$$\mathbb{E}_{\mathbb{P}_{real}(\mathbf{x})}\left[-\log\sum_{k=0}^{K+t-2}h_\theta(\mathbf{x})[k]\right] + \lambda_{\bar{\theta}}\mathbb{E}_{\mathbb{P}_{\bar{\theta}}(\mathbf{x}')}\left[-\log\sum_{k=K+t-1}^{K+t-2+F}h_\theta(\mathbf{x})[k]\right] +$$

$$\mathbb{E}_{\mathbb{P}_{real}(\mathbf{x},y)}\left[-\log h_\theta(\mathbf{x})[\sigma(y)]\right] + \mu_{\bar{\theta}}\mathbb{E}_{\mathbb{P}_{real}(\mathbf{x})}\left[-\log\sum_{k=K+t-1}^{K+t-2+F}h_\theta(\mathbf{x})[k]\right] \tag{31}$$

Note that $h_\theta(\mathbf{x})[i] \geq 0$ for all $i = 1, 2, \ldots, K+t-2+F$, and then we get

$$\mathbb{E}_{\mathbb{P}_{real}(\mathbf{x})}\left[-\log\sum_{k=0}^{K+t-2}h_\theta(\mathbf{x})[k]\right] \leq \mathbb{E}_{\mathbb{P}_{real}(\mathbf{x},y)}\left[-\log h_\theta(\mathbf{x})[\sigma(y)]\right], \tag{32}$$

$$\mathbb{E}\left[-\log\sum_{k=K+t-1}^{K+t-2+F}h_\theta(\mathbf{x})[k]\right] \leq \mathbb{E}\left[-\log\max_{k\in\{K+t-1,\ldots,K+t-2+F\}}h_\theta(\mathbf{x})[k]\right]. \tag{33}$$

Hence, Eq. 31 is upper bounded by

$$2\mathbb{E}_{\mathbb{P}_{real}(\mathbf{x},y)}\left[-\log h_\theta(\mathbf{x})[\sigma(y)]\right] + \lambda_{\bar{\theta}}\mathbb{E}_{\mathbb{P}_{\bar{\theta}}(\mathbf{x}')}\left[-\log\max_{k\in\{K+t-1,\ldots,K+t-2+F\}}h_\theta(\mathbf{x})[k]\right] +$$

$$\mu_{\bar{\theta}}\mathbb{E}_{\mathbb{P}_{real}(\mathbf{x})}\left[-\log\max_{k\in\{K+t-1,\ldots,K+t-2+F\}}h_\theta(\mathbf{x})[k]\right]. \tag{34}$$

Taking Eq. 34 as the optimization objective, we expand the expansion phase in original BEEF into a more stable form with multiple backward and forward prototypes, namely the expansion phase in Stable-BEEF.

Here we further discuss how we achieve the expanded fusion phase in Stable-BEEF. We also assume that we have trained a unifying model $h_{\theta_o}$ for all the prior tasks and there is a $\sigma_o$ maps labels to output index of $h_{\theta_o}$. Considering that those $t-1$ backward prototypes measures the confidence for $t-1$ prior tasks, similar to Eq. 10, we define

$$E_{\{\theta_o,\theta\}}(\mathbf{x},y) = \begin{cases} -\log\{h_{\theta_o}(\mathbf{x})[\sigma(y)] + \alpha_i h_\theta(\mathbf{x})[\sigma(y)] + \beta_i\}, & y \in \mathcal{Y}_i, i = 1, 2, \ldots, t-1 \\ -\log h_\theta(\mathbf{x})[\sigma(y)], & y \in \mathcal{Y}_t \end{cases}. \tag{35}$$

Then we have

$$\mathbb{P}_{\{\theta,\theta_o\}}(y|\mathbf{x}) = \begin{cases} \frac{h_{\theta_o}(\mathbf{x})[\sigma(y)]+\alpha h_\theta(\mathbf{x})[0]+\beta}{\sum_{m=1}^{M}[h_{\theta_o}(\mathbf{x})[m]+\alpha h_\theta(\mathbf{x})[0]+\beta]+\sum_{k=1}^{K}h_\theta(\mathbf{x})[k]}, & y \in \mathcal{Y}_o \\ \frac{h_\theta(\mathbf{x})[\sigma(y)]}{\sum_{m=1}^{M}[h_{\theta_o}(\mathbf{x})[m]+\alpha h_\theta(\mathbf{x})[0]+\beta]+\sum_{k=1}^{K}h_\theta(\mathbf{x})[k]}, & y \in \mathcal{Y}_n \end{cases}. \tag{36}$$

$\alpha_i$ and $\beta_i$ are obtained through the minimization of the negative log-likelihood on the exemplar-set.

## C   MORE EXPERIMENTAL SETTINGS

### C.1   EXEMPLAR SELECTION.

As we claimed, all the prior methods assume that all the old samples are available when selecting exemplars. Typically, they apply the exemplar selection strategy proposed in Rebuffi et al. (2017), where exemplars are carefully selected by greedily minimizing the derivation of the feature center between the selected exemplars and all the old samples. Besides, they assume all the old categories are equal and store the same number of exemplars for each old class. This violates the truth in many application scenarios, where available exemplars are usually imbalanced and even some instances for old categories become unavailable at the following sessions. Therefore, the robustness to the imbalance or lack of some categories is important for CIL methods. Assuming there are $k$ old classes

and we want to store $m$ exemplars for each old class. We design three protocols for the imbalance of exemplar-set: (1) **half-half**: in this protocol, half of $k$ classes are allowed to store more than $m$ exemplars and the other half of $k$ classes are only allowed to store less than $m$ exemplars. Specifically, defining the balance factor $\gamma \in [0, 1]$, half of $k$ classes store $(1 + \gamma)m$ exemplars while the other store $(1 - \gamma)m$ exemplars. (2) **exp**: in this protocol, we use a negative exponential sequence of length $k$ as the weight for each category. Specifically, the $i^{th}$ class has the weight $\exp(-\gamma i)$ and its number of exemplars is defined by $\left\lfloor \frac{\exp(-\gamma i)}{\sum_{j=1}^{k} \exp(-\gamma j)} \cdot km \right\rfloor$. (3) **random**: we uniformly sample $km$ exemplars from all available old instances. This protocol is the most similar to the original setting since the expectation of the number of exemplars for each category is $m$ statistically.

## C.2 Implementation details.

For ImageNet, we adopt the standard ResNet-18 (He et al., 2016) as our feature extractor and set the batch size as 256. The learning rate starts from 0.1 and gradually decays at milestones (170 epochs in total). For CIFAR-100, we use ResNet-32 (He et al., 2016) as our feature extractor and set the batch size to 128. The learning rate also starts from 0.1 and gradually decays at milestones (170 epochs in total). For both ImageNet and CIFAR-100, we use SGD with the momentum of 0.9 and the weight decay of 5e-4 at the expansion phase. At the fusion phase, we use SGD with a momentum of 0.9 and set the weight decay to 0, and train the fused model on the exemplar-set for 60 epochs. We apply the data augmenation following Rebuffi et al. (2017) and Wang et al. (2022a).To generating samples from $\mathbb{P}'_\theta(\mathbf{x})$ when learning the energy manifold, we use representations of real instances as the starting point of SGLD (Welling & Teh, 2011) following Contrastive Divergence (Hinton, 2002), which can be seen as a disturbance to the original representation of the real samples. In order to further accelerate the sampling process, we employed additional representation perturbation strategies, including mixup (Zhang et al., 2018), rotation, *etc.*, on the feature representations of samples. Therefore, we are able to generate new samples rather efficiently when learning the energy manifold. To further stabilize the training process and improve the performance, we expand the original BEEF into Stable-BEEF with multiple backward and forward prototypes. The number of backward prototypes is set to be the number of old tasks, and the number of forward prototypes is set to a constant number $F$. The detailed illustration and proof are provided in Appendix B. Besides, to simplify the training procedure and avoid the intractability of normalizing constants, we use a trade-off coefficient $\lambda$ to replace $\lambda_{\bar{\theta}}$ and $\mu_{\bar{\theta}}$ in our implementation.

# D More Discussions

## D.1 Contributions of BEEF

Our contribution is three-fold: 1) We propose a novel training paradigm for CIL: We efficiently train a specific module for the current task while achieving bi-directional compatibility and then fuse it with the prior model under minimal effort. 2) We provide a theoretical proof showing that the training cost is inherently the modeling of an energy-based model 3) We achieve state-of-the-art performance with lower training cost and maintain the performance when only randomly sampled old data is available while other methods fail dramatically.

## D.2 Difference and connections with prior methods

Here we discuss the crucial difference and connections between our method and prior works. Though Wang et al. (2021) has proposed to use an energy-based extra dimension to model the open world uncertainty in OOD detection, it is not suitable for CIL where old categories should be considered to alleviate forgetting. Besides, we take the joint distribution $\mathbb{P}(\mathbf{x}, y) = \mathbb{P}(\mathbf{x})\mathbb{P}(y|\mathbf{x})$ into consideration, which forces the model to learn discrimination ($\mathbb{P}(y|\mathbf{x})$) and be sensitive to the input distribution shift ($\mathbb{P}(\mathbf{x})$) and thus modules that do not match the input distribution will capture the distribution shift and weaken their influence on the ultimate predictions. Similar to Joseph et al. (2022), we also utilize the energy-based model to achieve energy alignment at the evaluation phase. While our energy-based model is inherently built at the training phase, Joseph et al. (2022) need to train an extra energy aligner after the training phase, introducing additional training costs. Apart from that, despite they assume that the task-ids of given samples are known in their implementation, we do not require that. Zhou et al. (2022) cleverly utilize multiple virtual prototypes to reserve feature space for future

unseen categories, thus achieving forward compatibility. Our method can be seen as a bi-directional compatible method, with one prototype $\mathbf{p}_b$ measuring the confidence for old categories and the other prototype $\mathbf{p}_f$ modeling the out-of-distribution probability.

### D.3 THE ROBUSTNESS OF BEEF TO THE IMBALANCED EXEMPLAR-SET

Here we give an explanation about why BEEF demonstrates strong robustness to the imbalance of exemplar-set. Previous methods, whether based on dynamic-structure or regularization, rely on a unifying feature representation and classifier. When the exemplar-set is unbalanced, that is, when the number of samples in some categories is very small or does not exist, the feature representation and classifier of the class are damaged. Although our method still needs to fuse all modules into a unified classifier, each module is decoupled from each other and is responsible for different tasks. Thus, when training new tasks, even if samples of some old classes are not available, the old modules responsible for these classes are not affected. In addition, when training new modules, we classify all old classes into a special backward prototype. This special backward prototype aims to learn the characteristics shared by categories in old tasks, which is insensitive to the imbalance of old samples. In the model fusion phase, the confidence of backward prototype is uniformly added to the output of old modules, thus acting as a soft task discriminator without affecting the predictions of old modules on old tasks.

### D.4 THE EFFICIENCY OF BEEF

**Training Cost Analysis.** Previous methods, whether based on knowledge distillation or dynamic structure, require the forward propagation of old modules when learning new tasks, to achieve a unifying classifier. Besides, with the increasing number of modules retained, dynamic-structure-based methods suffer from increasing training costs. The training of BEEF consists of two phases, expansion and fusion. At the expansion phase, we only need to train the newly created module independently without the involvement of prior modules, so that the training cost at the expansion phase is equivalent to tuning a single module and will not increase at incremental stages. Though we follow Stochastic Gradient Langevin Dynamics (SGLD) (Welling & Teh, 2011) to generate samples from $\mathbb{P}'_\theta(\mathbf{x})$ when learning the energy manifold. However, we have made many strategies to simplify and accelerate the process, including Contrastive Divergence, mixup, rotation, *etc*. At the fusion phase, we only need to learn two factors including scale factor $\alpha$ and bias factor $\beta$ through training on the randomly selected exemplar-set, and therefore the training cost is minimal compared to the expansion phase.

Particularly, we compare the training cost of BEEF with the methods including the classical dynamic-structure-based method DER (Yan et al., 2021) and regularization-based methods like iCaRL (Rebuffi et al., 2017) and WA (Wu et al., 2019). Without loss of generality, we take the $t^{th}$ incremental stage as an example. Although all the old modules are frozen in DER, the training process still requires the forward propagation of all $t-1$ old modules and the forward propagation of the new module and then applies backpropagation to update the new module. Apart from that, DER also needs to be the final classifier with a balanced reserved dataset. Therefore, the training process of DER at $t^{th}$ incremental stage is: $t \times$ forward propagation, 1 x backpropagation, $1 \times$ finetune classifier. Similarly, due to the requirements of knowledge distillation in regularization-based methods like iCaRL, the training cost is: $2 \times$ forward propagation, $1 \times$ backpropagation. In contrast, in BEEF, we first train the new module in a decoupled manner, which only requires: $1 \times$ forward propagation, $1 \times$ backward propagation. Then, the fusion phase only needs to tune two parameters with a small subset, whose cost is minimal and is even more efficient than the process of simply tuning the final classifier in DER. Therefore, the training cost of BEEF consists of: $1 \times$ forward propagation, $1 \times$ backward propagation, $1 \times$ finetune $\alpha$ and $\beta$.

Specifically, we compare the training time of BiC (regularization-based), DER (dynamic-structure-based) and BEEF at each incremental session on CIFAR-100 B0 10 steps protocol, the results are shown in Table. 4. We can observe that the average training time of BEEF is lower than both the regularization-based method BiC and the dynamic-structure-based method DER.

**Memory usage analysis.** Here, we report the peak memory for storing the exemplars and the learnable & frozen network parameters during the model training through all phases on B0 5 steps protocol of Benchmark CIFAR-100 and ImageNet-100 in Table. 5. For the model parameters, we use the type float32 to save all of them. That is, each parameter in the model takes 4 bytes of memory.

| Methods | Training time cost in each session (min) | | | | | | | | | | Average |
|---|---|---|---|---|---|---|---|---|---|---|---|
| | 1 | 2 | 3 | 4 | 5 | 6 | 7 | 8 | 9 | 10 | |
| BiC | 25 | 42 | 43 | 44 | 42 | 43 | 45 | 43 | 46 | 48 | 42.1 |
| DER | 25 | 37 | 43 | 49 | 53 | 58 | 66 | 74 | 78 | 91 | 57.4 |
| BEEF | 30 | 31 | 33 | 40 | 38 | 41 | 43 | 42 | 45 | 46 | 38.9 |

Table 4: Training time cost comparisons.

For images with width $W$ and height $H$, since each image has three channels and each pixel (with value 0-255) takes one byte, therefore each image takes $3 \times W \times H$ bytes of memory. Assuming that we have $N$ exemplars and the number of model parameters is $M$, the memory usage is calculated as $3 \times W \times H \times N + 4 \times M$. Note that the memory usage of RMM varies with different choice of configs. For CIFAR-100, the memory usage of it varies between 9.66 MB and 24.2 MB. For ImageNet-100, the memory usage of it varies between 378 MB to 1949 MB. We report the mean value of them in the above table.

| Benchmarks | Peak memory usage of different methods (MB) | | | | | | | | |
|---|---|---|---|---|---|---|---|---|---|
| | Replay | iCaRL | BiC | PodNet | FOSTER | DyTox | DER | RMM | BEEF |
| CIFAR-100 | 7.64 | 7.64 | 7.64 | 7.64 | 7.64 | **50.62** | 16.74 | 16.93 | 16.74 |
| ImageNet-100 | 330 | 330 | 330 | 330 | 330 | 333 | 554 | **1164** | 554 |

Table 5: Peak memory usage comparisons.

# E   MORE EXPERIMENTAL RESULTS

## E.1   DETAILED PERFORMANCE ON STANDARD CIFAR-100 B50 PROTOCOLS.

We report the detailed accuracy on standard CIFAR-100 B50 protocols with 5, 10, 25 incremental sessions in Fig. 7. We can observe that BEEF

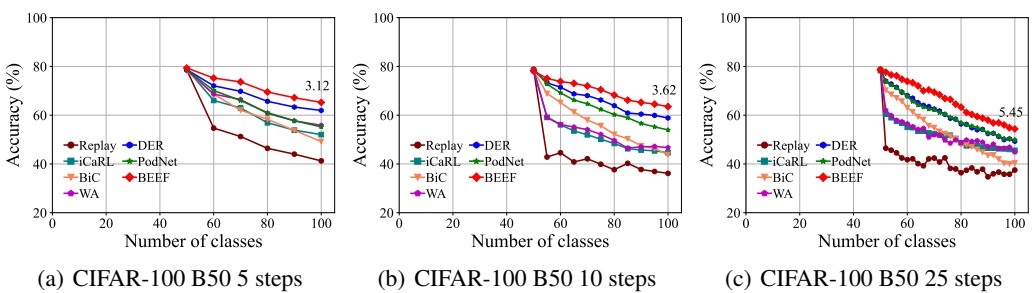

(a) CIFAR-100 B50 5 steps     (b) CIFAR-100 B50 10 steps     (c) CIFAR-100 B50 25 steps

Figure 7: Performance on standard CIFAR-100 B50 protocols.

## E.2   PERFORMANCE OF BEEF-COMPRESS WITH BACKBONES IN VARIOUS SIZES.

We report the detailed accuracy of BEEF-Compress with different backbones, including ResNet 20, ResNet 32, ResNet 44, ResNet 56, and ResNet 110 in Table 6. We can see that the training strategy of BEEF is consistently effective in backbones of various sizes. The accuracy of each incremental session has an upward trend as the model becomes larger.

## E.3   PERFORMANCE OF PRIOR WORK WITH FORWARD COMPATIBILITY.

Our overall training strategy is difficult to integrate with other existing CIL methods, but we find that our proposed forward compatibility implementation can be integrated into many methods. We experimented on the baseline regularization-based method WA and found that by implementing

| Backbones | Accuracy in each session (%) | | | | | | Average |
|---|---|---|---|---|---|---|---|
| | 1 | 2 | 3 | 4 | 5 | 6 | |
| ResNet 20 | 77.18 | 74.15 | 70.96 | 65.45 | 61.28 | 58.22 | 67.87 |
| ResNet 32 | 80.9 | 77.73 | 73.69 | 68.22 | 64.61 | 61.76 | 71.15 |
| ResNet 44 | 81.54 | 77.87 | 74.10 | 69.42 | 65.67 | 62.06 | 71.78 |
| ResNet 56 | 82.44 | 78.45 | 75.20 | 70.05 | 65.16 | 62.77 | 72.34 |
| ResNet 110 | 83.6 | 79.72 | 75.47 | 70.44 | 65.89 | 63.52 | 73.11 |

Table 6: Performance of BEEF with backbones in various sizes.

forward compatibility, that is, by classifying the perturbed features sampled by the built-in energy model into forward prototypes, we improve the performance by an evident margin. This might indicate that through achieving forward compatibility, our model is more likely to learn a representation with stronger generalization and transferability. We validate on CIFAR-100 B0 10 steps protocols with ResNet-18, and here we report the detailed accuracy at each incremental session in Table. 7, where WA$^+$ means WA with forward compatibility.

| Methods | Accuracy in each session (%) | | | | | | | | | | Average |
|---|---|---|---|---|---|---|---|---|---|---|---|
| | 1 | 2 | 3 | 4 | 5 | 6 | 7 | 8 | 9 | 10 | |
| WA | 89.5 | 78.85 | 75.73 | 71.4 | 68.22 | 64.05 | 62.2 | 57.62 | 55.3 | 53.53 | 67.64 |
| WA$^+$ | 90.8 | 80.65 | 76.9 | 71.97 | 69.22 | 65.12 | 63.39 | 58.74 | 56.74 | 55.24 | 68.87 |

Table 7: Performance of WA w/o the forward compatibility.

### E.4 Detailed performance on standard ImageNet-100 protocols.

We report the detailed accuracy on standard ImageNet-100 in Fig. 8. We can observe that BEEF achieves competitive performance under different ImageNet-100 protocols.

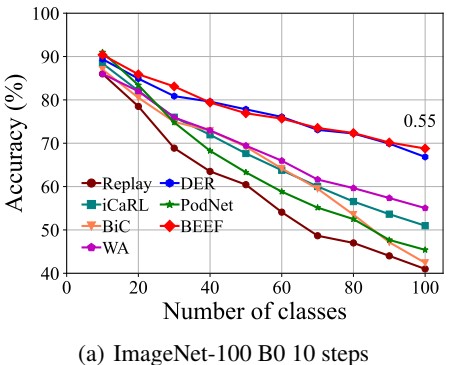

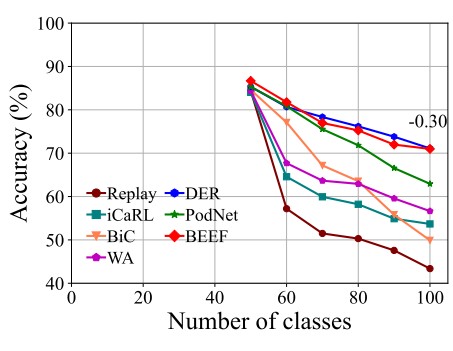

(a) ImageNet-100 B0 10 steps

(b) ImageNet-100 B50 5 steps

Figure 8: Performance on ImageNet-100.

### E.5 More results on imbalanced protocols.

We report more results on imbalanced protocols in Fig. 9. From the figure, we can observe that even under the configuration of a more balanced exemplar-set compared to that of Fig. 5, other methods still suffer from significant performance degradation, while BEEF achieves a much higher average accuracy than the other methods. Specifically, in the case of half-half and $\gamma = 0.5$, BEEF outperforms the best method by 4.25%, and in the case of exp and $\gamma = 0.9$, BEEF outperforms the best method by 9.80%. These results further demonstrate the robustness of our method BEEF in terms of exemplar-set selection strategy.

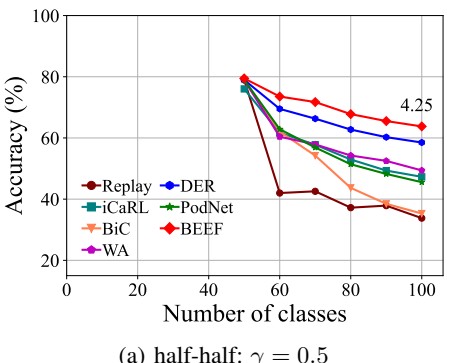 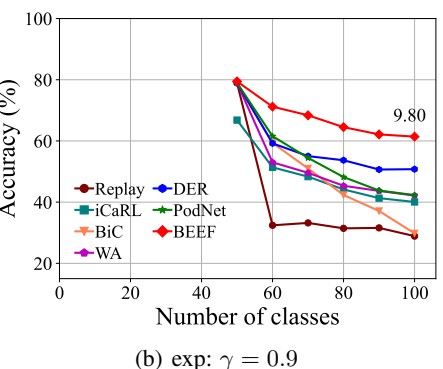

(a) half-half: $\gamma = 0.5$          (b) exp: $\gamma = 0.9$

Figure 9: Performance on imbalanced protocols.

