# OpenReview forum: " BEEF: Bi-Compatible Class-Incremental Learning via Energy-Based Expansion and Fusion"
_ICLR.cc/2023/Conference — ICLR 2023 poster_

### Official Review · Reviewer_zPVs · 2022-10-21

**Confidence:** 3
**Correctness:** 3
**Technical Novelty And Significance:** 3
**Empirical Novelty And Significance:** 3
**Recommendation:** 6

**Clarity, Quality, Novelty And Reproducibility:**

The paper is in general well writen and easy to follow, while I will also suggest the authors to give more explanation for their figure 2 (Learning energy manifold and test-time energy alignment)

Besides, for the learning of EBM, I think many important works of learning EBM as a generative model has not been included here, these include:

[1] A Theory of Generative ConvNet ICML 2016

[2] On learning non-convergent non-persistent short-run MCMC toward energy-based model. NeurIPS, 2019

[3] Cooperative learning of descriptor and generator networks. PAMI, 2020

[4] VAEBM: A symbiosis between variational autoencoders and energy-based models. ICLR, 2021

[5] Learning Energy-Based Generative Models via Coarse-to-Fine Expanding and Sampling. ICLR, 2021

[6] Learning energy-based model with variational autoencoder as amortized sampler. AAAI 2021

[7] A Tale of Two Flows: Cooperative Learning of Langevin Flow and Normalizing Flow Toward Energy-Based Model. ICLR, 2022


**Strength And Weaknesses:**

Strength:
1. I think the idea that applies the energy based model (EBM) to class incremental learning (CIL) is interesting;
2. The designs of the energy expansion objective and energy fusion function is meaningful;
3. The experimental results seem to support their claim that the new proposed framework works better.

Possible concerns:
1. In the training objective (Eq,7), the constant $\mu_{\theta}$ and $\lambda_{\theta}$ are actually still intractable and related to the normalizing constant of EBM. And the way the authors solve this problem is simple treating it as a hyper-parameter to tune. Theoretically this may be incorrect. And in practice, introducing a new hyperparameter may mean that the performance will depends on the setting/tuning of this new hyperparameter. If we look at Figure 6 b, we can see that the setting of $\lambda$ do influence the performance.

2. Also there are other parameters like $\alpha$ and $\beta$ that need to be tuned, too. The authors tune these parameters on a tiny  sub-dataset. I would like to know how sensitive the performance is to these two parameters settings. Also, whether tuning on a tiny sub-dataset is feasible in practice and can give accurate enough results.

3. For the energy-based model alignment part, the authors mention that they use Adam to accumulate gradient. Then I'm wondering whether they still include the noise term $\omega_i$ in eq 13 or just do gradient desent or ascent. And how would the aligned images looks like comparing to the original images.

**Summary Of The Paper:**

In this paper the authors propose to use energy-based model to improve the performance of Class-incremental learning. They try to make their model to be bi-directional compatible, i.e. forward compatible, which aims to enable old modules to sensitively capture the input distribution shift and backward compatible, which aims to enable the discrimination ability of old modules unaffected by new ones.Given data from new classes, a new energy-based model is first trained with a designed gradient equivalent objective function to avoid the intractable normalizing constant during the energy-based model expansion phase. And after that the new energy model is fused with the previous models with a designed fusing function. Through experimental results on CIFAR-100 and ImageNet-100, the authors demonstrate that their model work better than the baselines.

**Summary Of The Review:**

In summary, I like the idea proposed in this paper that applies the energy-based model theory to the Class-Incremental Learning task. And the results seems to support their claims, while on the other hand, I also have some concerns that I may want authors to further clarify.

---

> ### Author Response · Authors · 2022-11-19
> **Response to zPVs**
>
> We really appreciate your detailed, positive, and encouraging review.  We appreciate your valuable comments, and address your questions as follows:
>
> **1. Use the hyperparameter to replace the intractable normalizing constant.**
>
> By setting the intractable $\mu_\theta$ and $\lambda_\theta$ as hyperparameter $\lambda$, we not only simplify the implementation but also achieve better trade-off in balancing between the classification loss and the energy-based loss for uncertainty modeling. See Fig 6(b), we have conducted experiments with various values of $\lambda$, including 0, 1e-1, 1e-2, 1e-3 and the experimental results show that performance with different $\lambda$ fluctuates, but in a rather small change (at most 0.5%).
>
> **2.**   **How sensitive of the tuning parameters & whether the subset is feasible**
>
> As we claimed above the Equation. (12), the subset used to finetune parameters $\alpha$ and $\beta$ is exactly the exemplar-set we restored. And as we described in Section  4  CIFAR-100/ImageNet Protocol, the size of the exemplar-set is quite small, typically no more than 2000 exemplars. All the methods compared with 3EF are rehearsal based [1], that is, all those methods require a exemplar-set to help them better maintain old discrimination ability. Therefore, the use of the subset to tune the two hyperparameters is feasible.  Actually, many prior works use the exemplar-set to tune some parts of the model (especially the final fully-connected layer used as the classifier), such as DER [2], BiC [3], EEIL [4], etc.
>
> And as shown in Fig.5, Fig.8, 3EF maintains its performance when the exemplar-set is randomly selected and even of extreme imbalance (exemplars of half old classes are missing). ,  The two factors (i.e., $\alpha$ and $\beta$) are also tuned on the imbalanced exemplar-set. In Appendix. C. 4, we give an explanation of reasons why 3EF is robust to the imbalanced exemplar-set.
>
> As we described in Section 3.3, the introduction of $\alpha$ and $\beta$ aims to achieve a balance between the new module and old ones. Therefore, if the $\alpha$ and $\beta$ are not well-tuned, the ultimate prediction will have a strong tendency to either old classes or new ones, thus corrupting the accuracy.
>
> We initialize $\alpha$ and $\beta$ as 1 and 0, and we report the accuracy before and after tuning $\alpha$ and $\beta$ at each incremental stage on CIFAR-100 B50 5 steps protocol.
>
> |        | 2     | 3     | 4     | 5     | 6     |
> | ------ | ----- | ----- | ----- | ----- | ----- |
> | Before | 76.77 | 71.41 | 67.89 | 65.17 | 62.20 |
> | After  | 76.3  | 72.81 | 69.34 | 66.46 | 64.54 |
>
> **3.**   **Do we include the noise? How do the images look like?**
>
> We do include the noise. Since we use the embeddings of real instances as the starting point for the generation process (see Appendix C.2 Implementation Details), therefore the introduction of noises is necessary to generate different adversarial samples. As we described, we use the standard deviation of 1e-3 in either the sampling process or the test-time energy alignment process.
>
> Since we implement the SGLD in the hidden layers instead of the original image to facilitate the sampling process, therefore the running of SGLD does not influence the original image (see Appendix C.2 Implementation Details and Figure. 6(a)).
>
> **4.**  **Lack of important works of energy-based learning**
>
> Thank you for your suggestion, we have added brief discussions about these great works in Section 2 (Related Work).
>
>
>
> [1] Incremental Classifier and Representation Learning. CVPR2017
>
> [2] DER: Dynamically Expandable Representation for Class Incremental Learning. CVPR2021
>
> [3] Large Scale Incremental Learning. CVPR2019
>
> [4] End-to-end incremental learning. ECCV2018

---

> > ### Comment · Reviewer_zPVs · 2022-12-07
> > **I may want to keep my original rating**
> >
> > I would like to thank the authors for their detailed response. Currently I want to keep my rating as borderline accept.
> >
> > On one hand, the rebuttal does answer some of my questions, but I still hold concerns regarding the hyperparameter tuning in this work. For me, this proposed model contains several new hyper-parameters to be tuned and I think they are important to the performance of the model. Take $\lambda$ as an example, in Figure 6b, although the absolute value of the accuracy seems to be similar, if we consider the delta performance between totally removing the corresponding lost term $(\lambda = 0)$ and include this term with different $\lambda > 0$, this delta brought buy introducing this lost term is greatly influenced by $\lambda$.
> >
> > On the other hand, concerns raised by other reviewers also make sense to me. First, I would like to hear the opnion from so2y and 2BsG on whether they think the newly added explanation and comparison with other methods like FOSTER is convincing and fair enough to justify the model. And for the effectiveness of 3EF, I think it might be more straight-forward to compare different method directly in time like second or minutes instead of calculating the forward and backward times. After all, different method might have different time for running one iteration, especially when the new proposed model includes SGLD during training, which might take more time then other methods.
> >
> > In all, currently, I still lean to accept this paper, but the remaining concerns prevents me from further raise my rate. Thus, I want to keep my rating to borderline accept.

---

> > > ### Author Response · Authors · 2022-12-07
> > > **Thank you for your responese**
> > >
> > > Thank you so much for your comment.
> > >
> > > For the first question, there is a gap in performance when lambda = 0 and lambda > 0. We argue that this actually illustrates the effectiveness of our proposed `forward compatibility`. When lamda = 0, it means that 3EF will no longer classify the samples sampled by the energy model to classify into the forward prototype, that is, no longer achieve forward compatibility.
> > >
> > > In addition, regarding the specific running time, we have reported the specific running time of each stage of different methods in Table 4 in the revision. We compare 3EF with both the dynamic structure-based method DER and the regularization-based method BIC. From the table, we can see that the average training time of 3EF is less than these two methods.
> > >
> > > Hope this clarifies your doubts.

---

> > ### Comment · Reviewer_zPVs · 2022-12-12
> > **Related works**
> >
> > After checking the paragraph about energy-based learning in the related works section, I find the reference cited is wrong for the statement “Xiao et al. (2021) propose to learn a VAE to initialize the finitestep MCMC for efficient amortized sampling of the EBM.”  That one should have been the reference [6] (Xie et al 2021,  Learning energy-based model with variational autoencoder as amortized sampler) in the list I provided. It is good to see the authors complete the EBM related work by involving different EBM variants and EBM applications of EBMs, but it looks like most of the EBM application references are from the field of machine learning. Actually, there is a lot of prior work about EBM applications in the field of computer vision. I suggest the authors consider them.
> >
> > (1) EBM for video modeling and generation
> >
> > Synthesizing Dynamic Pattern by Spatial-Temporal Generative ConvNet. CVPR 2017.
> >
> > Learning Energy-based Spatial-Temporal Generative ConvNet for Dynamic Patterns. PAMI 2019
> >
> > (2) EBM for 3D  volumetric shape modeling and generation
> >
> > Learning Descriptor Networks for 3D Shape Synthesis and Analysis. CVPR 2018.
> >
> > Generative VoxelNet: Learning Energy-Based Models for 3D Shape Synthesis and Analysis. PAMI 2020.
> >
> > (3) EBM for 3D unordered point cloud modeling and generation.
> >
> > Generative PointNet: Deep Energy-Based Learning on Unordered Point Sets for 3D Generation, Reconstruction and Classification. CVPR 2021.
> >
> > (4) EBM for internal learning
> >
> > Patchwise Generative ConvNet: Training Energy-Based Models from a Single Natural Image for Internal Learning. CVPR 2021
> >
> > (5) EBM for inverse  optimal control and autonomous driving
> >
> > Energy-Based Continuous Inverse Optimal Control. IEEE Transactions on Neural Networks and Learning Systems (TNNLS) 2022
> >
> > (6) EBM for unpaired image-to-image translation
> >
> > Learning Cycle-Consistent Cooperative Networks via Alternating MCMC Teaching for Unsupervised Cross-Domain Translation. AAAI 2021.
> >
> > (7) EBM for supervised image-to-image translation / conditional learning
> >
> > Cooperative Training of Fast Thinking Initializer and Slow Thinking Solver for Conditional Learning. PAMI 2021.
> >
> > (8) EBM for salient object prediction
> >
> > Energy-Based Generative Cooperative Saliency Prediction. AAAI 2022
> >
> > (9) EBM for graph generation
> >
> > GraphEBM: Molecular Graph Generation with Energy-Based Models. ArXiv 2021

---

> > > ### Author Response · Authors · 2022-12-12
> > > **Thank your for your response!**
> > >
> > > Thank you very much for pointing out inappropriate information in our citations, which will be corrected in the final version.
> > >
> > > And we really appreciate your affirmation of our work, saying that we "complete the EBM-related work by involving different EBM variants and applications of EBMs".
> > >
> > > Apart from that, thank you very much for introducing these great works on EBM-related computer vision applications, and we are happy to add a discussion of these works in the final version.

---

### Official Review · Reviewer_so2y · 2022-10-23

**Confidence:** 4
**Correctness:** 3
**Technical Novelty And Significance:** 2
**Empirical Novelty And Significance:** 2
**Recommendation:** 5

**Clarity, Quality, Novelty And Reproducibility:**

Fair: The paper is somewhat clear, but some important details are missing or unclear.

**Strength And Weaknesses:**

Strength:
This paper proposed to train a specific module for new samples and then fuse it with the prior model. Based on theoretical inspiration from the energy-based model, the model performance is further improved.

Weaknesses:
1. This paper proposes a method so-called ‘Efficient’, where training new models do not require the forward propagation of old models for knowledge distillation. However, SGLD in EBM indeed increased the computation. Different modules are required in different steps, resulting in more parameters amount. Some quantitative analysis may be more helpful to understand. (eg. parameters amount)
2. Comparison with recent SOTA methods including RMM [1], and Foster [2] are missing. I suspect that maybe the paper cannot achieve true state-of-the-art performance. Eg. in Foster CIFAR100 B0 10 steps: 72.90. But this paper achieved 71.94.
3. Some detailed accuracy plots at each incremental step on magnet benchmarks should be presented in this paper to show the effectiveness of the method.
4. This paper needs to be highly polished. Eg. section C.4 ‘. Our method can is a bi-directional compatible method,’


[1] RMM: Reinforced Memory Management for Class-Incremental Learning
[2] FOSTER: Feature Boosting and Compression for Class-Incremental Learning

**Summary Of The Paper:**

This paper studied the problem in dynamic-structure-based CIL methods, namely that coupled training of modules in different incremental steps leads to additional training costs and spoilage of eventual predictions. The authors proposed a unifying energy-based theory and framework called 3EF to train independent modules in a decoupled manner at each incremental step, and then fused the modules into a unifying classifier. Experimental results suggest that the proposed approach outperforms on the CIFAR100, ImageNet100, and ImageNet1000 benchmarks.

**Summary Of The Review:**

The proposed method is somewhat novel and applying the energy-based model to CIL community is not new. On the other hand, I have doubts about the experimental results. My initial evaluation of this paper is “weak reject”.

---

> ### Author Response · Authors · 2022-11-19
> **Response to so2y [1/2]**
>
> Thank you for your  kind review.  We appreciate your valuable comments, and address your questions as follows:
>
> **1. Why 3EF is Efficient?**
>
> We have added a detailed analysis of the training cost of different methods in **Appendix Section D.4** (refer to this section for more details).
>
> Previous methods, whether based on knowledge distillation or dynamic structure, require the forward propagation of old modules when learning new tasks, to achieve a unifying classifier. Besides, with the increasing number of modules retained, dynamic-structure-based methods suffer from increasing training costs.  As we claimed, the training of  3EF consists of two phases, expansion and fusion. At the expansion phase, we only need to train the newly created module independently without the involvement of prior modules, so that the training cost at the expansion phase is equivalent to tuning a single module and will not increase at incremental stages. Though we follow Stochastic Gradient Langevin Dynamics (SGLD) to generate samples from $\mathbb P^\prime_\theta(\mathbf x)$ when learning the energy manifold. However, instead of initializing samples randomly as the starting point, we use representations of real instances as the starting point of SGLD at the hidden layer, which can be seen as a disturbance to the original hidden representation of the real samples and greatly accelerate the generation process. Besides, we use the Adam optimizer to automatically adjust the step size and consider the cumulative gradient to further improve the generation efficiency. We run only 5 updates to get the generated samples.  At the fusion phase, we only need to learn two factors including scale factor $\alpha$ and bias factor $\beta$ through training on the randomly selected exemplar-set for about 30 epochs, whose training cost is minimal compared to the expansion phase.
>
> Without loss of generality, we take the $t^{th}$ incremental stage as an example.  Although all the old modules are frozen in DER, the training process still requires the forward propagation of all $t-1$ old modules and the forward propagation of the new module and then applies backpropagation to update the new module. Apart from that, DER also needs to be the final classifier with a balanced reserved dataset. Therefore, the training process of DER at $t^{th}$ incremental stage is: $t$ $\times$ forward propagation, 1 x backpropagation, $1$ $\times$ finetune classifier. Similarly, due to the requirements of knowledge distillation in regularization-based methods like iCaRL, the training cost is: $2$ $\times$ forward propagation, $1$ $\times$ backpropagation. In contrast, in 3EF, we first train the new module in a decoupled manner, which only requires: $1$ $\times$ forward propagation, $1$ $\times$ backward propagation. Then, the fusion phase only needs to tune two parameters with a small subset, whose cost is minimal and is even more efficient than the process of simply tuning the final classifier in DER. Therefore, the training cost of 3EF consists of: $1$ $\times$ forward propagation, $1$ $\times$ backward propagation, $1$ $\times$ finetune $\alpha$ and $\beta$.
>
> To  conclude:
>
> DER:  $t$ $\times$ forward propagation + $1$ x backpropagation + $1$ $\times$ finetune classifier
>
> iCaRL: $2$ $\times$ forward propagation + $1$ $\times$ backpropagation
>
> BiC: $2$ $\times$ forward propagation + $1$ $\times$ backpropagation + $1$ $\times$  finetune bias-correction layer
>
> 3EF: $1$ $\times$ forward propagation + $1$ $\times$ backpropagation + $1$ $\times$  finetune $\alpha$ and $\beta$
>
> We also report the detailed empirical training time comparison in Table 4, verifying the efficiency of 3EF, achieving the average training time 38.9 minutes (the average training time of BiC and DER is 42.1 minutes and 57.4 minutes respectively).

---

> ### Author Response · Authors · 2022-11-19
> **Response to so2y [2/2]**
>
> **2.**   **Comparison with RMM、FOSTER**
>
> We have added comparison with more CIL methods, including FOSTER [1], RMM [2], DyTox [3], and RPSNet [4], where FOSTER, DyTox, and RPSNet are dynamic-structure-based methods (see Table 1, Table 2). FOSTER additionally uses AutoAugmentation to enhance the sample efficiency and improve classification accuracy. RMM achieves a better memory management strategy. With the same training memory, RMM chooses some new samples to train and discards the rest to allow restoring more old exemplars. Note that all these are orthogonal to the method itself, and therefore we combine the augmentation and memory management strategy with 3EF-Distill for a fair comparison with these methods, which we regard as 3EF-Distill++.   As shown in Table 1, and Table 2, 3EF/3EF-Distill++ can achieve state-of-the-art performance on most protocols without careful hyperparameter tuning.
>
> **3.**   **Performance on ImageNet**
>
> Thank you for your suggestion,  we have added the detailed accuracy plots at each incremental step on ImageNet in Appendix Section E.3.
>
> **4.**   **Need to be highly polished**
>
> Thank you for the suggestion, we have carefully polished the paper.
>
> [1] Feature Boosting and Compression for Class-incremental Learning. ECCV 2022
>
> [2] RMM: Reinforced Memory Management for Class-Incremental Learning. NeurIPS2021
>
> [3] DyTox: Transformers for Continual Learning with DYnamic TOken eXpansion. CVPR 2022
>
> [4] An adaptive random path selection approach for incremental learning. ArXiv 2019

---

> ### Author Response · Authors · 2022-12-09
> **Deadline for discussion approaching**
>
> Dear reviewer so2y,
>
> Hope this message finds you well.  We have tried our best to enrich the experiment part and show the effectiveness of 3EF. As the end of the discussion is approaching,  we have had a further discussion with the other reviewers and addressed concerns.  Are there any other questions about our work and the comments? we are happy to engage in a discussion addressing them.

---

### Official Review · Reviewer_2BsG · 2022-10-24

**Confidence:** 4
**Correctness:** 3
**Technical Novelty And Significance:** 2
**Empirical Novelty And Significance:** 2
**Recommendation:** 6

**Clarity, Quality, Novelty And Reproducibility:**

The quality of this paper is high in terms of writing. Clarity is high.
Novelty is poor.
Reproducibility is uncertain as no code is given.

**Strength And Weaknesses:**

Strength******

1. The paper is clearly written and easy to follow. Though there are many formulations of entropies, the interpretation of each formula is quite clear and straightforward. It is easy to follow.
2. Implementation details are well-presented in the paper. There is a compressive analysis of the model's sensitivity to hyperparameters.
3. The model fusion and alignment modules are interesting and novel.

Weaknesses*******

The main idea of this paper, "using independent models for different learning phases", is often taken as a baseline. See the "independent" setting in table 3 of this paper [Gradient Episodic Memory for Continual Learning, NIPS 2017]. It is straightforward and not fair to the models using a single model.

Though this submission designs a novel module of model fusion and alignment based on entropy, its poor performance is still a concern, given it has used so many copies of backbone networks (i.e., a much higher number of network parameters compared to the related methods using a single model). For example, its performances on ImageNet-based datasets (all settings in ImageNet-100 and -1000) are consistently lower than those of FOSTER [1] -- where FOSTER is a more efficient model than DER, in terms of computational speed and parameter quantity.

[1] FOSTER: Feature Boosting and Compression for Class-Incremental Learning. ECCV 2022.


**Summary Of The Paper:**

This paper aims to learn dynamic-structure-based CIL models in a decoupled manner. It learns independent models for different tasks and then fuses them at a low cost to make phase-wise predictions. The idea is simple and seems effective compared to simple baseline methods such as ICARL and PODNet, but can not achieve the top performance compared to the recent work FOSTER (the citation is missing from this ICLR submission)---which is a more effective version of DER.

**Summary Of The Review:**

My main concerns are two-fold. 1) The unfair comparison between this paper and related works of using a single model for CIL. 2) The poor performance compared to an unmentioned sota method FOSTER.

---

> ### Author Response · Authors · 2022-11-19
> **Response to 2BsG**
>
> Thank you for your review.  We appreciate your valuable comments, and address your questions as follows:
>
> **1.**   **The idea is used and the novelty is poor**
>
> First, the idea of “the independent models for different phases” in GEM [1] is mainly about **task-incremental learning**, where the task labels of given samples are **known** at the evaluation phase. While 3EF aims to solve the problem of **class-incremental learning**, where the task labels of given samples are **unknown** at the evaluation phase [2]. In 3EF, instead of directly using the task label to choose the relevant module to make the decision (since we do not know the task label in class-incremental learning), 3EF automatically finds the input-distribution-related module and increases its impact on the final predictions. Therefore, the mechanism of 3EF is completely different from the “independent” in GEM.
>
> Besides, we hope to clarify that though we do express the idea of “using independent models for different learning phases” in 3EF, our core contribution and novelty are nothing about that.  Our core contribution, as we claimed (see Appendix Section D.1, D.2, D.3, D.4)
>
> - In 3EF, we propose a unifying energy-based expansion and fusion theory and framework, to our knowledge, no prior works have ever done that. Most prior works in class-incremental learning are empirically inspired and lack a solid theoretical framework.
> - We propose to achieve bi-directional compatibility when learning isolated models to s alleviate possible conflicts of different models. Through achieving bi-directional compatibility, the fused model can automatically decide which module should take the major position in the final predictions.
> - We greatly improve the efficiency of the training of dynamic-structure-based methods through a two-stage learning process, including expansion and fusion (see Table 4).
> - We extend the class-incremental learning with rehearsal to a more difficult scenario, where old exemplars are randomly selected and even the labels of old exemplars are not required. 3EF maintains its performance when other methods fail dramatically.
>
> **2.**   **The unfair comparison with single model methods and lower performance than FOSTER**
>
>  In our original version of 3EF, we aim to propose a compatible expansion and fusion strategy in class-incremental learning. Since 3EF retains all the old modules expanded, its memory usage of model parameters is N times as the single-backbone methods. However, we show that the common compress strategy such as pruning or distillation can also be combined with 3EF.
>
> For a fair comparison, we apply the same data augmentation (i.e., AutoAugmentation) and use the same knowledge distillation proposed in FOSTER to compress the fused dual-branch model into a single skeleton in 3EF at each incremental session, which we call 3EF-Distill++. In this way, the memory usage of model parameters after each incremental session keeps consistent with the other methods.
>
> We report the performance of FOSTER and 3EF-Distill++ in Table 1 and Table 2. We can see that even though the distillation strategy in FOSTER is not specially designed for 3EF,  the average accuracy of 3EF-Distill++ outperforms FOSTER in most protocols, showing the great potential of 3EF.
>
> Apart from the accuracy performance on standard settings, 3EF has many advantages in nature compared with FOSTER:
>
> - 3EF can handle a randomly selected, class-imbalanced exemplar-set and maintain its performance while FOSTER cannot (we provide an explanation of reasons why those methods fail in Appendix Section D.3).
> - Our ablation study shows that 3EF is more robust to the choice of hyperparameters than FOSTER. In FOSTER, the choice of $\beta_1$ in FOSTER greatly influences the trade-off between old and new classes. As we experimented, simply changing the $\beta_1$ from $0.95$ to $0.94$ led to 1.35 average accuracy decline. In contrast, the ablation studies show that the performance of 3EF with different $\lambda$ fluctuates, but in a rather small change (at most 0.5%, see Fig 6(b)).
> - The training of 3EF/3EF-Distill even does not requires any old labels (since we use only one backward prototype to cluster features of all old instances), which means that 3EF can achieve competitive performance under the “semi-supervised” setting (i.e., only class supervision on the new task is available),
>
> [1] Gradient Episodic Memory for Continual Learning. NIPS 2017
>
> [2] Three scenarios for continual learning. arXiv 2019

---

### Official Review · Reviewer_hrbR · 2022-10-24

**Confidence:** 4
**Correctness:** 3
**Technical Novelty And Significance:** 3
**Empirical Novelty And Significance:** 3
**Recommendation:** 6

**Clarity, Quality, Novelty And Reproducibility:**

### Clarity & Quality

The paper is well-written and easy to follow.

### Novelty

The energy-based method for CIL seems interesting. However, we still need to validate that the proposed method doesn’t violate the memory usage protocol in CIL.

### Reproducibility

The authors didn’t provide the open-source code.


**Strength And Weaknesses:**

### Strengths

- Extensive experiment results are provided.
- This paper is well-organized and easy to follow.
- The proposed method is technically sound

### Weaknesses

- The memory usage of the model parameters might violate the class-incremental learning benchmark protocol. The authors propose to train independent modules and then integrate them. However, each module requires a memory budget to save them. We can also use the same memory budget to save more exemplars to improve performance. Based on this, the authors should provide ablation results showing saving the independent modules is more efficient compared to saving the exemplars.

- There is no analysis of memory usage. As the independent modules also need to use the memory, it is important to provide the memory usage in Tables 1 and 2.

- The authors claimed that their method aims to address the “training cost” issue in CIL. However, there is no analysis of the training cost. The authors need to provide a comparison of the training cost to show their method performs better than other baselines.

- There is no open-source code. The authors didn’t provide the implementation in the submission. It is important to provide the following researchers with enough materials to reproduce the results.


**Summary Of The Paper:**

In this paper, the authors propose a new CIL method, which trains independent modules in a decoupled manner and integrates the modules into a unifying classifier. They provide an energy-based explanation for the proposed method with theoretical analyses. Extensive experiment results on CIFAR-100, ImageNet-100, and ImageNet-1000 are provided.

**Summary Of The Review:**

Overall, this is an interesting paper. However, there is no open-source code. Besides, the analyses of memory usage and training cost are not provided. As these two points are very essential to the final contributions, the overall score is borderline reject.

---

> ### Author Response · Authors · 2022-11-19
> **Response to hrbR**
>
> Thanks for your appreciation of our writing and experiments. We appreciate your comments and address your concerns as follows:
>
> **1&2. Memory usage of the model parameters might violate the protocol & Lack of Analysis of memory usage.**
>
> Thanks for pointing this out. In our original version of 3EF, we aim to propose a compatible expansion and fusion strategy in class-incremental learning. Since 3EF retains all the old modules expanded, its memory usage of model parameters is N times as the single-backbone methods. However, we show that the common compress strategy such as pruning or distillation can also be combined with 3EF.
>
> For a fair comparison, we apply the same data augmentation and use the same knowledge distillation proposed in FOSTER to compress the fused dual-branch model into a single skeleton in 3EF at each incremental session, which we call 3EF-Distill++. From Table 1, and Table 2, we can see that even if the distillation strategy in FOSTER is not specially designed for 3EF, 3EF-Distill++ still shows very competitive performance, which indicates the great potential of 3EF.
>
> In this way, the memory usage of model parameters after each incremental session keeps unchanged, except for the final classifier for classification.
>
> **3.**   **Analysis of training cost**
>
>
> We have added a detailed analysis of the training cost of different methods in **Appendix Section D.4** (refer to this section for more details).
>
> Without loss of generality, we take the $t^{th}$ incremental stage as an example.  Although all the old modules are frozen in DER, the training process still requires the forward propagation of all $t-1$ old modules and the forward propagation of the new module and then applies backpropagation to update the new module. Apart from that, DER also needs to be the final classifier with a balanced reserved dataset. Therefore, the training process of DER at $t^{th}$ incremental stage is: $t$ $\times$ forward propagation, 1 x backpropagation, $1$ $\times$ finetune classifier. Similarly, due to the requirements of knowledge distillation in regularization-based methods like iCaRL, the training cost is: $2$ $\times$ forward propagation, $1$ $\times$ backpropagation. In contrast, in 3EF, we first train the new module in a decoupled manner, which only requires: $1$ $\times$ forward propagation, $1$ $\times$ backward propagation. Then, the fusion phase only needs to tune two parameters with a small subset, whose cost is minimal and is even more efficient than the process of simply tuning the final classifier in DER. Therefore, the training cost of 3EF consists of: $1$ $\times$ forward propagation, $1$ $\times$ backward propagation, $1$ $\times$ finetune $\alpha$ and $\beta$.
>
> To  conclude:
>
> DER:  $t$ $\times$ forward propagation + $1$ x backpropagation + $1$ $\times$ finetune classifier
>
> iCaRL: $2$ $\times$ forward propagation + $1$ $\times$ backpropagation
>
> BiC: $2$ $\times$ forward propagation + $1$ $\times$ backpropagation + $1$ $\times$  finetune bias-correction layer
>
> 3EF: $1$ $\times$ forward propagation + $1$ $\times$ backpropagation + $1$ $\times$  finetune $\alpha$ and $\beta$
>
> We also report the detailed empirical training time comparison in Table 4, verifying the efficiency of 3EF, achieving the average training time 38.9 minutes (the average training time of BiC and DER is 42.1 minutes and 57.4 minutes respectively).
>
> **4.**  **There is no open-source code**
>
> we will open-source our implementation after being accepted.

---

> ### Comment · Reviewer_hrbR · 2022-12-07
> **Thanks for the feedback from the authors.**
>
> Thanks for the feedback from the authors.
>
> **For my Q1**: I think the "3EF-Distill++" setting is reasonable. The authors addressed my concerns.
>
> **For my Q2**: the authors didn't add memory usage to Tables 1 and 2. I think memory usage is very important and needs to be included in the tables.
>
> **For my Q3**: As the authors claimed that their method aims to address the “training cost” issue in CIL, the training cost analysis should appear in the main paper.
>
> **For my Q4**: the authors addressed my concerns.
>
> I will upgrade my rating if the authors can address my Q2 and Q3.

---

> > ### Author Response · Authors · 2022-12-08
> > **Thank you for your response!**
> >
> > Thank you very much for your comment.
> >
> > 1. **About the memory usage**, we argue that we have well solved the issue of extra memory usage through achieving 3EF-Distill++. Here, we report the peak memory for storing the exemplars and the learnable & frozen network parameters during the model training through all phases on B0 5 steps protocol of Benchmark CIFAR-100 and ImageNet-100.
> >
> > | methods\benchmarks | CIFAR-100   | ImageNet-100 |
> > | ------------------ | ----------- | ------------ |
> > | Replay             | 7.64 MB     | 330MB        |
> > | iCaRL              | 7.64MB      | 330MB        |
> > | BiC                | 7.64MB      | 330MB        |
> > | PodNet             | 7.64MB      | 330MB        |
> > | FOSTER             | 7.64MB      | 330MB        |
> > | Dytox              | **50.62MB**     | 333MB        |
> > | DER                | 16.74MB     | 554MB        |
> > | RMM                | 16.93MB | **1164MB**   |
> > | 3EF                | 16.74 MB    | 554MB        |
> > | 3EF-Distill++      | 7.64MB      | 330MB        |
> >
> > > Note that the memory usage of RMM varies with different choice of configs. For CIFAR-100, the memory usage of it varies between 9.66MB and 24.2 MB. For ImageNet-100, the memory usage of it varies between 378MB to 1949 MB.  We report the mean value of them in the above table.
> >
> > Considering that now we are unable to revise the paper, we will add this memory usage comparison in our final version.
> >
> > 2.**About the analysis of training cost**, we have already provide a detailed analysis of training cost **in Appendix Section C.4**.  In the paper, We show that 3EF only requires 1 $\times$ forward propagation + 1 $\times$ backward propagation and 1 $\times$  finetune $\alpha$ and $\beta$. We also provide a specific training time comparison on CIFAR-100 B0 10 steps protocols and empirically verify the training efficiency of 3EF in Table 4.
> >
> > | methods | 1    | 2    | 3    | 4    | 5    | 6    | 7    | 8    | 9    | 10   | Average |
> > | ------- | ---- | ---- | ---- | ---- | ---- | ---- | ---- | ---- | ---- | ---- | ------- |
> > | BIC     | 25   | 42   | 43   | 44   | 42   | 43   | 45   | 43   | 46   | 48   | 42.1    |
> > | DER     | 25   | 37   | 43   | 49   | 53   | 58   | 66   | 74   | 78   | 91   | 57.4    |
> > | 3EF     | 30   | 31   | 33   | 40   | 38   | 41   | 43   | 42   | 45   | 46   | 38.9    |
> >
> >
> >
> > Hope this address your concerns.

---

> > > ### Comment · Reviewer_hrbR · 2022-12-08
> > > **Thanks for the further feedback from the authors.**
> > >
> > > Thanks for the further feedback from the authors!
> > >
> > > The answers above partially addressed my concerns in Q2 and Q3. For memory usage, it is very important to include how these numbers are calculated. Are you using BMP or JPEG images? Are you using float32 or float64 to save the parameters? This information needs to be included in the revision.
> > >
> > > I upgraded my rating to "6: marginally above the acceptance threshold".

---

> > > > ### Author Response · Authors · 2022-12-09
> > > > **Thank you for your response!**
> > > >
> > > > Thank you so much for your kind reply and for upgrading the rating.
> > > >
> > > > For the model parameters, we use the type float32 to save all of them. That is, each parameter in the model takes 4 bytes of memory.
> > > >
> > > > For images with width $W$ and height $H$, since each image has three channels and each pixel (with value 0-255) takes one byte, therefore each image takes $3\times H\times W$ bytes of memory.
> > > >
> > > > Assuming that we have $N$ exemplars and the number of model parameters is $M$, the memory usage is calculated as $3\times H\times W\times N + 4 \times M$.
> > > >
> > > > Thanks for the precious suggestion, we will add these discussions in the final version.
> > > >
> > > > Thank you again for reconsidering our work.

---

### Official Review · Reviewer_6aJg · 2022-10-27

**Confidence:** 3
**Clarity, Quality, Novelty And Reproducibility:** This work is well motivated and well …
**Correctness:** 3
**Technical Novelty And Significance:** 3
**Empirical Novelty And Significance:** 3
**Recommendation:** 6

**Strength And Weaknesses:**

Pros:
1. This work is well-motivated by the training efficiency of dynamic-structure-based methods and possible conflict among modules in addressing novel classes.
2. A novel method from the perspective of energy-based methods is proposed. The proposed energy-based framework consists of two stages, i.e., expansion phase where new modules are trained to satisfy both backward and forward compatibility and fusion phase where a unified classifier for all seen categories is obtained.
3. The proposed method also improves robustness to the imbalance or lack of some categories in exemplar-set.
4. This work is well-written and the structure of the paper is well-organized. The symbols, equations and their descriptions are clear and easy to follow.
5. The experiment setting is clearly explained. Extensive experiments, including both comparison with SOTAs and ablation studies, demonstrate the effectiveness of the proposed method.


Cons:
1. In this work only one dynamic-structure-based method is compared with. More strong dynamic-structure-based methods with rehearsal are suggested to compare with.
2. Performance of applying the proposed training strategy to more existing models.
3. 'For benchmark ImageNet-100 ...' in Sec. 4.1 are suggested to put in another paragraph for clarity.

**Summary Of The Paper:**

In this work authors propose a novel dynamic-structure-based method for class incremental learning from the perspective of energy-based models. The proposed Efficient Energy-based Expanasion and Fusion (3EF) framework introduces the concept of bi-directional compatibility and decouples the training of different modules with a expansion and fusion two-stage training strategy. In addition, authors also show less requirement on the selection of  exemplar-set for rehearsal. Extensive experiments on three popular benchmarks show that the proposed method performs favorately against state-of-the-art methods.

**Summary Of The Review:**

Overall, I think this work is well prepared and would inspire and advance the development of increamental learning field.

---

> ### Author Response · Authors · 2022-11-19
> **Response to 6aJg**
>
> Thank you for your positive and encouraging review. We appreciate your valuable comments, and address your questions as follows:
>
> **1.**   **More strong dynamic-structure-based methods with rehearsal should be compared.**
>
> We have added comparison with more CIL methods, including FOSTER [1], RMM [2], DyTox [3], and RPSNet [4], where FOSTER, DyTox, and RPSNet are dynamic-structure-based methods. As shown in Table 1, and Table 2, 3EF/3EF-Distill++ achieves state-of-the-art performance on most protocols.
>
> **2.**   **Performance of applying the proposed training strategy to more existing models.**
>
> we conduct experiments of 3EF-Distill  on CIFAR-100 B50 5 steps protocol with backbones in different sizes. We report the detailed accuracy in each session, and results (see Table 5) show that the framework of 3EF is consistently effective and the accuracy of each incremental session has an upward trend as the model becomes larger.
>
> **3.**   **Put the Benchmark of ImageNet-100 in another paragraph**
>
> Thanks for your suggestion, we have made the revision in 3EF.
>
> [1] Feature Boosting and Compression for Class-incremental Learning. ECCV 2022
>
> [2] RMM: Reinforced Memory Management for Class-Incremental Learning. NeurIPS2021
>
> [3] DyTox: Transformers for Continual Learning with DYnamic TOken eXpansion. CVPR 2022
>
> [4] An adaptive random path selection approach for incremental learning. ArXiv 2019

---

### Author Response · Authors · 2022-11-19
**General Response**

We thank all the reviewers for their precious comments. We acknowledge that all reviewers do catch the shining point of the paper, saying our work is well-written (6aJg, hrbR, 2BsG), interesting and meaningful (zPVs,), and easy to follow (6aJg, hrbR).

In this rebuttal, we have given careful thought to the reviewers’ suggestions and made the following revisions to our manuscript to answer the questions and concerns:

- In **Section 2 (Related Work)**, we add discussions about more important works about EBM, especially the works recommended by reviewer (zPVs).
- In **Section 3 (3EF-DISTILL)**, we argue that common compression strategies are orthogonal with 3EF and can be directly combined with the training framework of 3EF. We show that with the same compression strategy used in FOSTER [1], 3EF can still achieve competitive results while maintaining the single skeleton, which well solves the issue of increasing the memory usage of models.
- In **Section 4 (Empirical Studies)**, we compare with more CIL methods, including FOSTER [1], RMM [2], DyTox [3], and RPSNet [4].  Considering that these methods use stronger data augmentation or a more clever memory management strategy to increase performance, we show that all these techniques are orthogonal to 3EF and can be combined into **3EF/3EF-Distill** directly. We compare the original version of **3EF** and **3EF-Distill++** (which uses the same compression strategy in FOSTER to retain a single skeleton and augmentation in FOSTER to enhance the sample efficiency) with the other methods (see Table 1, Table 2).   Due to limited GPU resources, we are **still running** 3EF-Distill++ experiments on ImageNet. We will report the results **in the form of post** in openreview.
- In **Appendix Section D.4**, we add a more detailed analysis of the training cost of different methods, including regularization-based methods,  dynamic-structure-based methods, and 3EF. We also provide a concrete running time comparison among BiC [5] (regularization-based), DER [6] (dynamic-structure-based), and 3EF,  and the experimental results show that the growth of the training time of 3EF is slower than that of BiC and DER.
- In **Appendix Section E.2**, we conduct experiments of 3EF-Distill  on CIFAR-100 B50 5 steps protocol with backbones in different sizes. We report the detailed accuracy in each session, and results (see Table 5 ) show that the framework of 3EF is consistently effective and the accuracy of each incremental session has an upward trend as the model becomes larger.
- In **Appendix Section E.3**, we report the detailed accuracy of 3EF on benchmark ImageNet-100.
- we will open-source our implementation after being accepted.

We have highlighted the revised part in our manuscript in **blue** color. Please check the answers to specific comments.

[1] Feature Boosting and Compression for Class-incremental Learning. ECCV 2022

[2] RMM: Reinforced Memory Management for Class-Incremental Learning. NeurIPS2021

[3] DyTox: Transformers for Continual Learning with DYnamic TOken eXpansion. CVPR 2022

[4] An adaptive random path selection approach for incremental learning. ArXiv 2019

[5] Large Scale Incremental Learning. CVPR2019

[6] DER: Dynamically Expandable Representation for Class Incremental Learning. CVPR2021

---

> ### Author Response · Authors · 2022-11-24
> **The performance of 3EF-Distill++ on Benchmark ImageNet-100**
>
> Here we report the performance of 3EF-Distill++ on Benchmark ImageNet-100
>
> | Protocols               | Avg Top1 | Avg Top5 | Last Top1 | Last Top5 |
> | ----------------------- | -------- | -------- | --------- | --------- |
> | ImageNet100 B0 10 steps | 79.34    | 93.30    | 71.12     | 88.94     |
> | ImageNet100 B50 5 steps | 80.52    | 94.10    | 74.62     | 91.42     |
>
> We observe that 3EF-Distill++ achieves state-of-the-art performance on ImageNet-100 (both top1 and top5 accuracy).
>
> Specifically, the detailed performance at each incremental session:
>
> **ImageNet-100 B0 10 steps**
>
> | ImageNet-100 B0 10 steps | 1 | 2 | 3 | 4 | 5 | 6 | 7 | 8 | 9 | 10 |
> | ----------------------- | -------- | -------- | --------- | --------- | --------- | --------- | --------- | --------- | --------- | --------- |
> | Top1 | 89.8 | 86.0 | 83.0 | 81.14 |80.16|78.07|76.17|74.82|72.84|71.12|
> | Top5 | 99.2 | 97.0 | 95.84 | 94.70 |93.76|92.17|90.91|90.82|89.64|88.94|
>
> **ImageNet-100 B50 5 steps**
>
> | ImageNet-100 B50 5 steps | 1     | 2     | 3     | 4     | 5     | 6     |
> | ------------------------ | ----- | ----- | ----- | ----- | ----- | ----- |
> | Top1                     | 86.48 | 85.0  | 81.46 | 79.05 | 76.56 | 74.62 |
> | Top5                     | 96.52 | 95.58 | 95.06 | 93.58 | 92.20 | 91.42 |

---

### Author Response · Authors · 2022-12-07
**Deadline for discussion approaching**

Dear reviewers,

As the end of the discussion is approaching, we would like to thank you again for the valuable discussion and feedback. We have tried our best to enrich the experiment part and show the effectiveness of 3EF. Are there any other questions about the contributions of this paper? If you have any comments, concerns, or questions left, we are happy to engage in a discussion addressing them.

---

### Decision · Program_Chairs · 2023-01-20

**Decision:**

Accept: poster

**Justification For Why Not Higher Score:**

All reviewers are willing to weakly accept the paper, so the AC doesn't want to go beyond the poster.

**Justification For Why Not Lower Score:**

All reviewers agree that the paper is novel, interesting and technically sound, and has sufficient experiments and good empirical results. The paper is worthy to publish.

**Metareview: Summary, Strengths And Weaknesses:**

This paper proposes a novel dynamic-structure-based method for class incremental learning from the perspective of energy-based models. The proposed Efficient Energy-based Expansion and Fusion (3EF) framework trains independent modules in a decoupled manner while achieving bi-directional compatibility among modules through two additionally allocated prototypes, and then integrating them into a unifying classifier with minimal cost. Extensive experiments conducted on three popular benchmarks show that the proposed method can obtain state-of-the-art performance. All reviewers agree that the paper is novel, interesting and technically sound, and has sufficient experiments and good empirical results. The rebuttal has successfully addressed the major concerns raised by the reviewers, and the paper has been improved by completing the related works and including new empirical results. Thus, the AC recommends accepting the paper.


**Note From Pc:**

if the above contains the word "oral" or "spotlight" please see: "oral" presentation means -> notable-top-5% and "spotlight" means -> notable-top-25%. As stated in our emails, we are disassociating presentation type from AC recommendations